# TAD evolutionary and functional characterization reveals diversity in mammalian TAD boundary properties and function

Mariam Okhovat [1,15] ✉, Jake VanCampen[1,15], Kimberly A. Nevonen[1], Lana Harshman[2,3], Weiyu Li[2,3], Cora E. Layman[1], Samantha Ward[1], Jarod Herrera[1], Jackson Wells[1], Rory R. Sheng [2,3], Yafei Mao [4,14], Blaise Ndjamen[5], Ana C. Lima [6], Katinka A. Vigh-Conrad[6], Alexandra M. Stendahl [6], Ran Yang [6], Lev Fedorov[7], Ian R. Matthews[8], Sarah A. Easow [8], Dylan K. Chan[8], Taha A. Jan [9], Evan E. Eichler[4,10], Sandra Rugonyi [11], Donald F. Conrad [6,12], Nadav Ahituv [2,3] ✉ & Lucia Carbone [1,6,12,13] ✉

Topological associating domains (TADs) are self-interacting genomic units crucial for shaping gene regulation patterns. Despite their importance, the extent of their evolutionary conservation and its functional implications remain largely unknown. In this study, we generate Hi-C and ChIP-seq data and compare TAD organization across four primate and four rodent species and characterize the genetic and epigenetic properties of TAD boundaries in correspondence to their evolutionary conservation. We find 14% of all human TAD boundaries to be shared among all eight species (ultraconserved), while 15% are human-specific. Ultraconserved TAD boundaries have stronger insulation strength, CTCF binding, and enrichment of older retrotransposons compared to species-specific boundaries. CRISPR-Cas9 knockouts of an ultraconserved boundary in a mouse model lead to tissue-specific gene expression changes and morphological phenotypes. Deletion of a human-specific boundary near the autism-related *AUTS2* gene results in the upregulation of this gene in neurons. Overall, our study provides pertinent TAD boundary evolutionary conservation annotations and showcases the functional importance of TAD evolution.

The three-dimensional (3D) organization of the genome plays a fundamental role in orchestrating chromatin interactions that regulate gene expression and shape phenotypes[1,2]. Chromatin conformation capture assays, namely Hi-C, have shed light on 3D genome organization and revealed that chromosomes are compartmentalized into kilo- to mega-base scale segments termed topologically associated domains (TADs). TADs often contain gene(s) that can interact with regulatory elements located in the same domain, while interactions across domains are prevented by flanking elements that are often bound by specific proteins including CTCF, cohesin complexes and RNA-polymerase II[3]. These insulating elements are commonly known as TAD boundaries, and considering their significant role in defining functional interaction domains, they are thought to be critical for maintaining proper genome function.

Disruption of TAD boundaries has been associated with ectopic gene interactions, gene misregulation and aberrant phenotypes such as cancer[4], limb malformation[5] and neurodevelopmental disease[6]. As a result, the function and position of TAD boundaries in the genome are likely subject to some degree of evolutionary constraint and conservation. When TADs were first described[7–9], a direct comparison between human and mouse TAD organization revealed that a large portion of TAD boundaries was shared between the two species (i.e., 53.8% of human boundaries were also present in mouse)[7]. This finding was corroborated by several subsequent lines of direct and indirect evidence. For example, direct comparisons of the *Hox*[10] and *Six*[11] loci detected similar TAD structures across several species. Further convincing—albeit indirect—evidence of TAD conservation came from studies providing evidence for selection against structural variations (SVs) that disrupt genome topology; In the human population, evolutionary SVs were shown to be depleted at TAD boundaries, while this pattern was absent in patients with developmental delay[12]. In addition, in the highly rearranged gibbon genome[13], breakpoints of evolutionary chromosomal rearrangements between human and gibbon were found to be enriched at TAD boundaries, resulting in the preservation of TADs even when synteny is lost[14]. Findings from these and other studies have reinforced the notion that TAD boundaries remain unchanged during evolution due to deleterious consequences of TAD disruption[15]. However, most of the evidence supporting TAD conservation is indirect and/or only limited to a few species or loci. Moreover, a few recent studies present evidence refuting high conservation of TAD boundaries across species, even those as closely related as human and chimpanzee[16]. Hence, there is growing uncertainty around the level of evolutionary TAD conservation[17] and a need for further reassessment of TAD conservation via comprehensive direct cross-species comparisons.

Gene regulation plays a key role in species evolution[18]. Therefore, a comparative investigation of TAD boundaries and the gene regulatory domains they form can shed new light on mechanisms contributing to adaptations, speciation, and evolution. Regions of high TAD conservation represent crucial loci whose disruption may lead to reduced fitness[19], while regions with evolutionarily diverged TAD organization may be associated with gene interaction/expression changes that underlie evolutionary novelties across species. Indeed, differences in 3D genomic interactions in human-chimpanzee induced pluripotent stem cells (iPSC) have been associated with inter-specific differences in gene expression[16]. Furthermore, 3D chromatin structures identified in the fetal human cortex, but not in rhesus and mouse, may have contributed to human-specific features of brain development[20], highlighting the power of using a comparative approach in assessing TAD evolution and inferring TAD boundary functionality.

Here, in order to assess the level of evolutionary TAD conservation in mammals, we identified TAD boundaries across four primate and four rodent species and performed direct comparisons to group them based on their evolutionary conservation. Characterization of the genetic and epigenetic profiles of TAD boundary across evolutionary groups, as well as generation of in vivo and in vitro deletions, allowed us to investigate the function of these boundaries and their relevance to development, disease and evolution.

## Results

### Cross-species comparison of TAD boundaries reveals variation in evolutionary conservation

To identify TADs and their boundaries across species, we selected eight species from the primate and rodent orders and generated genome-wide chromatin conformation (Hi-C) maps (Fig. 1). The selected species all had high-quality reference genomes and spanned different evolutionary distances within their phylogenetic order. Furthermore, for both orders, we included species that

have experienced several evolutionary genomic rearrangements in a relatively short evolutionary time. For primates, in addition to human (representing great apes) and rhesus macaque (representing Old World Monkeys), we included two species of gibbons, as these small apes exhibit significantly rearranged genomes between genera despite having diverged only ~5 million years ago[13]. The two selected gibbon species have very different karyotypes and chromosome numbers (*Nomascus leucogenys*, 2n = 52 (Supplementary Note 1) and *Hylobates moloch*, 2n = 44), referred to as Nomascus and Hylobates hereon. From rodents, we selected mouse and rat, as well as two relatively closely related *Mus* species with divergent karyotypes (*M. caroli*, 2n = 40 and *M. pahari*, 2n = 48), referred to as Caroli and Pahari from hereon[21]. We used public Hi-C data for human lymphoblastoid cell line (LCL) (Supplementary Data 1) but generated new Hi-C data for all other species from one male and one female and merged the two replicates after confirming that they were highly correlated (r = 0.66–0.75) (Supplementary Data 1, Supplementary Fig. 1a). Merging the replicate Hi-C libraries resulted in an average of 440 ± 82 (mean ± stdev) million valid reads per species, with human having the highest (583 million valid reads) and rat having the lowest (338 million valid reads). Hi-C resolution was high (<10Kb) overall and estimated to be 6.9 ± 2.2 Kb across species, with Pahari having the highest (4.5 Kb) and rat having the lowest (9.9 Kb) resolution (Supplementary Data 1). We also generated CTCF and histone (H3K4me1, H3K27ac, H3K4me3 and H3K27me3) ChIP-seq data from the same samples (Supplementary Data 2). For rhesus and all rodent species, we used liver tissue. However, due to the challenge of sampling from humans and endangered gibbons, we used LCL from these species.

Using our Hi-C data, we identified an average of 7177 ± 481 (mean ± stdev) TADs across the genomes of all species, with mouse (N = 6542) and rhesus (N = 7845) having the lowest and highest numbers of predicted TADs, respectively (Supplementary Data 1). Across all species and tissues, TAD boundaries exhibited local enrichment of CTCF and two histone marks associated with active transcription, H3K27ac and H3K4me3 (Supplementary Fig. 1b). Over half of TAD boundaries (62.2% ± 5.5; mean ± stdev) overlapped genes in all species, with the highest and lowest percent overlap present in human (70.7%) and Pahari (56.4%), respectively. On average, less than half (40 ± 5%; mean ± stdev) of TAD boundaries in all species contained at least one CTCF binding site, with mouse having the highest (48%) and Nomascus having the lowest (36%) percentages (Supplementary Fig. 2).

### Genetic and epigenetic properties of boundaries vary depending on their evolutionary conservation

TADs have likely emerged at different evolutionary times and been exposed to different degrees of selective pressures. Hence, the evolutionary conservation and cross-species stability of TAD boundaries in a genome may vary greatly. In order to assess the evolutionary conservation of TAD boundary placement across species, boundaries from each species were lifted over to the human genome coordinates (hg38, Supplementary Data 3) and merged within 10Kb of each other to create a "union boundary" reference map. Boundaries from all species were then compared against each union boundary to determine their presence across species (Supplementary Data 4). Since our Hi-C data originated from two different tissues, patterns of boundary distribution across species may also be influenced by tissue differences[22]. To validate that the effect of tissue origin is weaker than the evolutionary relationship between samples, we investigated patterns of cross-species/tissue TAD boundary overlap and distribution. We found that boundaries were shared more often among closely related species, even when originating from different tissues (Supplementary Fig. 3). Also, the overall variation in the genomic distribution of boundaries was better explained by phylogenetic order (i.e., primate vs. rodent) rather than by tissue (Supplementary Note 2, Supplementary Figs. 1c and 3), highlighting that evolutionary distance

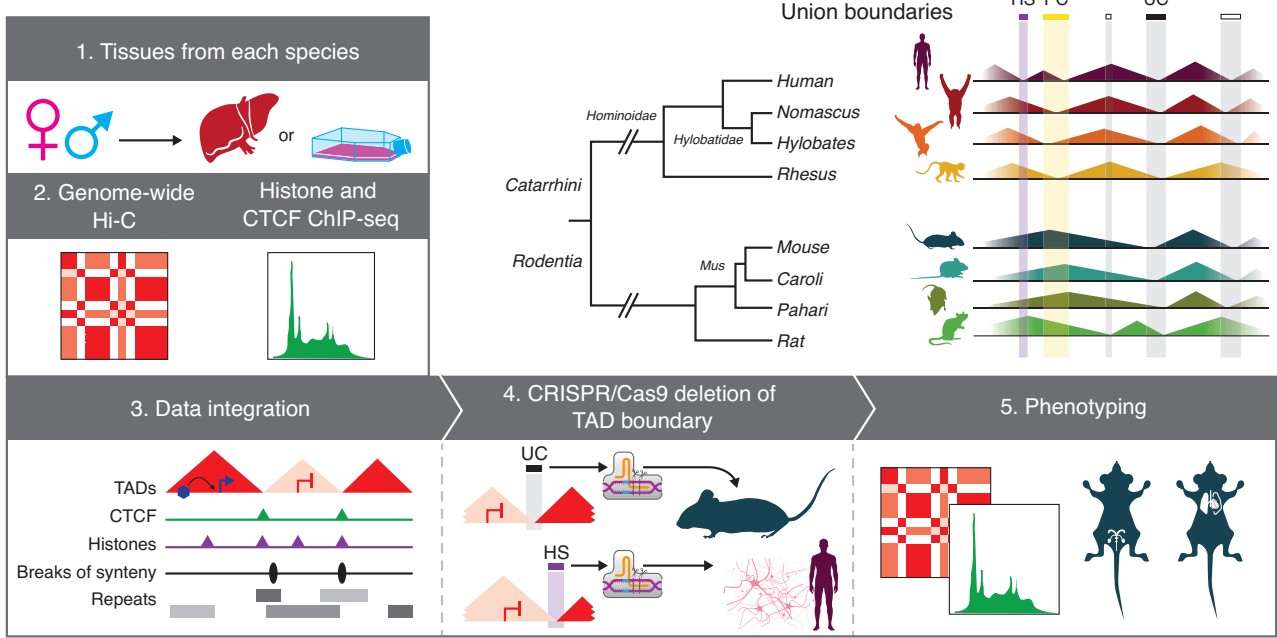

**Fig. 1 | Direct multi-species comparisons shed light on the evolutionary conservation of TAD boundaries.** Flowchart of the study design is shown, along with a scheme of the union boundary approach and our evolutionary criteria for annotating human-specific (HS), primate-conserved (PC) and ultraconserved (UC) TAD boundaries. The following artwork is licensed and modified from iStock.com/

Alex_Doubovitsky (liver icon), iStock.com/VectorMine (CRISPR icon), iStock.com/ Atlas Studio (human icon), iStock.com/Bullet_Chained (gibbon icons), iStock.com/ AlonzoDesign (macaque icon), iStock.com/Nedea (mouse, Caroli, Pahari icons), iStock.com/ -Userba9fe9ab_931 (rat icon).

has a stronger impact on shaping cross-species TAD organization among our samples.

We grouped boundaries present in the human and mouse reference based on the number of species sharing them (e.g., 1-way, 2-way, 3-way…., 8-way) (Supplementary Data 5) so that the genetic and epigenetic features of these groups could be compared within their respective genome. Comparison of the overall Hi-C interaction matrices revealed that interaction frequencies were overall lower across boundary groups that had higher cross-species conservation (Supplementary Fig. 4a). To better quantify this, we also estimated the insulation score[23] of TAD boundaries, which is a measure of interactions passing across each genomic region ("Methods"). We observed that as the level of conservation increased, the insulation scores decreased, indicating that boundaries shared by more species lead to higher separation between neighboring TADs (Supplementary Fig. 4b). In agreement with this finding, boundaries shared by more species overlapped CTCF binding sites and genes more frequently than less conserved ones (Supplementary Fig. 4c, d). Overall, these observations indicate that regardless of tissue and species, functional and structural properties of TAD boundaries differ depending on their cross-species conservation.

We then used the phylogenetic relationship between species to classify human and mouse TAD boundaries based on their evolutionary conservation and estimated age into the following groups: (1) ultraconserved boundaries (examples in Supplementary Fig. 5), which are shared across all species in this study and date back to at least the common ancestor of primates and rodents (~80 mya); (2) primate-conserved or rodent-conserved, which are only shared within each order and are likely at least ~25 mya and ~15 mya old respectively; and (3) species-specific, which are only present in the mouse (mouse-specific) or the human (human-specific) genome and represent younger boundaries that have recently evolved in each genome (~7 mya for human and ~3 mya for mouse). Furthermore, to exclude human boundaries that are shared with other great apes, we removed any human-specific boundary also found in Hi-C data from

chimpanzee and gorilla LCL[24]. Based on our classifications, 13.6% ($N = 1023$) of the boundaries in the human genome in LCL were ultraconserved, 6% ($N = 491$) were primate-conserved, and 15% ($N = 1130$) were human-specific. A roughly similar distribution was found in the mouse genome in liver, where 19.1% ($N = 1023$) of the TAD boundaries were ultraconserved, 2.1% ($N = 115$) were rodent-conserved, and 15.0% ($N = 807$) were mouse-specific (Supplementary Data 4 and 5).

We then used species-specific epigenetic and genetic data to investigate the properties of TAD boundaries as a function of their conservation in mouse and human genomes. We found that less conserved boundary groups in both human and mouse genomes showed significantly lower insulation scores and less overlap with genes compared to older and more conserved boundary groups (two-tail Wilcoxon rank sum test, $p < 0.001$; Fig. 2a, b). As differences in interaction patterns and insulation strength of TAD boundary groups could, at least partially, be attributed to their different frequency of overlap with genes, we also showed that insulation scores of both ultraconserved and species-specific TAD boundaries in mouse and human were significantly more extreme than randomly selected boundaries with the same frequency of overlap with genes (one-tail permutation $p$-value < 0.05) (Supplementary Note 3, Supplementary Fig. 6). The epigenetic landscape was also variable across boundary groups, with species-specific TAD boundaries in both human and mouse genomes showing higher enrichment of chromatin states associated with active transcription start sites, bivalent chromatin, and CTCF signal (Fig. 2c, Supplementary Note 4). Consistently, ultraconserved boundaries contained significantly more CTCF binding sites compared to order-conserved and species-specific boundaries in both the human and mouse genomes (two-tail Wilcoxon rank sum test, $p < 0.001$) (Fig. 2d). Hence, regardless of species and tissue, boundaries stratified based on evolutionary conservation differ in their genetic and epigenetic properties, with ultraconserved boundaries displaying stronger insulation and higher dependence on CTCF binding compared to species-specific boundaries.

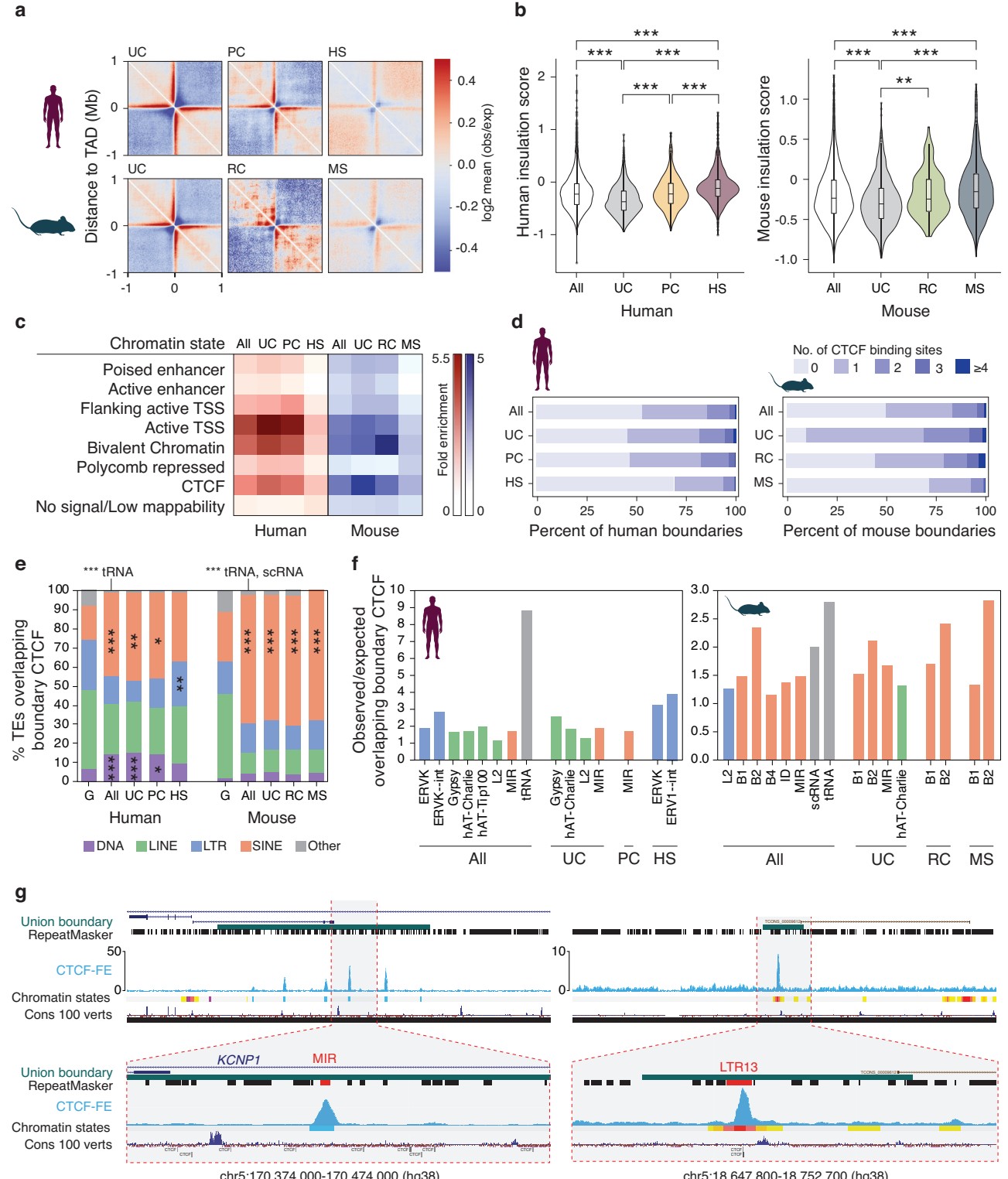

## Transposable elements contribute to the evolution of conserved and species-specific TAD boundaries

Transposable elements (TEs) can introduce CTCF binding sites upon their insertion in host genomes[25] and as such, their co-option may have contributed to the evolution of TAD boundaries[26,27]. Analysis of CTCF binding sites located inside TAD boundaries in each species showed that over half of them overlap TEs across species (ranging from 53 to 77%, depending on species; Supplementary Fig. 2). In the human genome, we found significant enrichment (binomial test; $p < 0.005$) of SINE and DNA

repeat classes at CTCF binding sites inside ultraconserved and primate-conserved boundaries (Fig. 2e). Among the enriched SINE repeats, the most noteworthy family was MIR, a family of ancient t-RNA-derived and highly conserved TEs[28,29]. MIR elements have been previously shown to provide insulation in the human genome in a CTCF-independent manner[30], however we found 14% of ultraconserved, 12% of primate-conserved and 10% of all human TAD boundaries to have at least one CTCF binding site overlapping a MIR element (Fig. 2f, g). In contrast to ultraconserved boundaries, CTCF sites within the human-specific TAD

**Fig. 2 | Genetic and epigenetic features of TAD boundaries vary as a function of their evolutionary conservation. a** Heatmaps of Hi-C interaction show differences in genomic interactions across evolutionary TAD boundary groups. **b** Violin plots show insulation score differences across evolutionary TAD boundary groups in human (left) and mouse (right). Number of boundaries included in each category in the human genome is as follows: $n_{All}$ = 7401, $n_{UC}$ = 1023, $n_{PC}$ = 491, and $n_{HS}$ = 1130. In the mouse genome, the number of boundaries are $n_{All}$ = 5354, $n_{UC}$ = 1023, $n_{RC}$ = 115 and $n_{MS}$ = 807. The ends and center lines of boxplots demonstrate the 1st, 2nd (median) and 3rd quartiles, while whiskers represent minimum and maximum insulation scores. *P*-values are based on two-sided Wilcoxon rank sum test and are only reported when <0.05. In human, $W_{HS\ vs.\ All}$ = 5167940, $p_{HS\ vs.\ All}$ < 2.2e-16; $W_{HS\ vs.\ PC}$ = 338547, $p$ = 2.133e-12; $W_{HS\ vs.\ UC}$ = 863127, $p$ < 2.2e-16; $W_{UC\ vs.\ All}$ = 2806392, $p$ < 2.2e-16; $W_{UC\ vs.\ PCl}$ = 179565, $p$ < 2.2e-16. In mouse, $W_{MS\ vs.\ All}$ = 2435686, $p$ = 5.045e-09; $W_{MS\ vs.\ UC}$ = 522658, $p$ < 2.2e-16; $W_{UC\ vs.\ All}$ = 2351016, $p$ = 6.821e-13; $W_{UC\ vs.\ RC}$ = 47850, $p$ = 0.001026. **c** Fold-enrichment of chromatin states at evolutionary TAD boundary groups in human (red) and mouse genome (blue). **d** Percent of TAD boundaries harboring CTCF binding sites across evolutionary groups. **e** Class composition of TEs overlapping CTCF binding sites in TAD boundaries is shown for boundaries in the human (left) and mouse genomes (right). Significantly over-represented TE classes based on a negative binomial test are marked with asterisks (\*\*\**p* < 0.0005, \*\**p* < 0.005, \**p* < 0.05). Low abundance TE classes are grouped as "Other", with only those that are significantly enriched mentioned on top. **f** Observed/Expected values for TE families significantly over-represented (*p* < 0.05) at TAD boundary CTCF binding sites. Bar colors represent TE class membership, based on colors used in panel (**e**). **g** UCSC genome browser screenshots of a MIR-derived CTCF binding site inside an ultraconserved boundary (left) and an ERV-derived CTCF binding site within human-specific TAD boundary (right). CTCF ChIP-seq fold-enrichment (CTCF-FE) track appears in blue. Hsap human, Mmus mouse, G genome-wide, All All TAD boundaries, US ultraconserved, PC primate-conserved, HS human-specific, RC rodent-conserved, MS mouse-specific; \*\*\**p* < 0.0005, \*\**p* < 0.005, \**p* < 0.05. No multiple comparisons adjustments were made.

boundaries were enriched in LTR elements, namely the ERV-K family (Fig. 2f, g), which is the most recently endogenized and transcriptionally active ERV family in the human genome[31]. However, these ERVs do not appear to be human-specific, indicating that the TE insertion event likely predates the establishment of the human-specific TAD boundaries at these loci. In the mouse genome, the SINE repeat class was significantly enriched (binomial test; *p* < 0.005) at CTCF binding sites of ultraconserved, rodent-conserved and mouse-specific boundaries. Similar to the human genome, among SINEs, the MIR family was significantly over-represented in CTCF binding sites of ultraconserved TAD boundaries, suggesting that this TE family has been involved in the evolution of TAD organization at least since the common ancestor of primates and rodents. Of note, in the mouse genome, the rodent-specific B1 and B2 repeat families were enriched at CTCF binding sites across all evolutionary boundary groups (Fig. 2f), supporting the previously proposed hypothesis that many CTCF sites in the mouse genome are derived from B elements[25]. Altogether, our data highlights the correlation between the evolutionary age of TAD boundaries and TEs, suggesting that waves of TE insertions followed by co-option have contributed to the spatial organization of the genome at different evolutionary time points.

## Species-specific boundaries are over-represented at evolutionary breaks of synteny

In the past, we and others[12,14,32] have shown that breakpoints of evolutionary chromosomal rearrangements are over-represented at TAD boundaries, and this has been interpreted as evidence that selection pressures maintain TAD integrity even when genome synteny is disrupted. Here, we took advantage of our high-resolution Hi-C data and improved genome assemblies[21,33] (Supplementary Note 1) to re-assess the extent of the co-occurrence between breakages of synteny (BOS) regions and TAD boundaries across species. BOS, which are the result of different types of chromosomal rearrangements (i.e., translocations, inversions, fissions), were identified by pairwise comparisons of non-human primate genomes against the human reference genome, whereas Caroli and Pahari genomes were compared to mouse ("Methods" and Supplementary Fig. 7). Consistent with previous reports[13,33,34], the highest number of evolutionary rearrangements was found in gibbon genomes (168 BOS in Hylobates and 130 in Nomascus) and the lowest number of BOS was found in Caroli (7 BOS relative to mouse; Supplementary Data 6).

We observed a strong reduction in genomic interaction frequency across BOS, as well as a notable dip in the collective insulation score at BOS in all three non-human primate species (Fig. 3a). Caroli and Pahari showed similar patterns, but the signal was localized and much weaker, likely due the relatively small number of BOS in these species. The overlap between BOS and TAD boundaries was statistically significant in both gibbons but not in the other species (Fisher's exact test *p* < 0.05; Fig. 3b and Supplementary Data 7).

The formation of TAD boundaries that overlap BOS regions could either predate or follow the evolutionary rearrangement event, but thus far, the relative timing of these events has not been investigated. Boundaries predating the rearrangement event are expected to have higher cross-species conservation compared to those established specifically in association with the breakage. We leveraged the large number of BOS in our study to investigate these scenarios. To do so, we collectively investigated cross-species conservation of TAD boundaries present near all BOS across species. To avoid underestimating conservation due to reduced synteny at BOS, we only considered boundaries that could successfully be lifted to the human reference genome (90 out of the 149 boundaries found near all BOS). We observed an over-representation of species-specific and low-conservation TAD boundaries near BOS (Fig. 3c). Next, we focused on the gibbon species, which were the only species with statistically significant boundary/BOS overlap. Previous studies focused on the Nomascus genome[35] indicate that roughly half of the rearrangements in this genus are shared with other gibbon genera, including Hylobates, and therefore likely occurred in the common gibbon ancestor, while the remaining half occurred after the genera split ~5 million years ago. Using this data, we assessed the evolutionary timing of BOS and determined that most BOS overlapping TAD boundaries in our study in both Nomascus and Hylobates (86% and 74%, respectively) were species-specific and likely occurred independently in the two genera (Supplementary Data 8). We also found that ultraconserved boundaries and primate-conserved boundaries were significantly underrepresented at BOS, while overlaps with species-specific boundaries were significantly more prevalent than expected (one-tailed permutation *p* < 0.05) (Fig. 3d) in both gibbon genomes. Overall, these findings suggest that the formation of many TAD boundaries near BOS has followed or co-occurred with the evolutionary chromosomal rearrangement event.

## Deletion of an ultraconserved TAD boundary results in tissue-specific functional phenotypes

TAD boundaries help prevent ectopic gene regulatory interactions across neighboring TADs, and their disruption can lead to gene misregulation[5,15]. As such, there may be an association between the TAD boundary conservation level and the nature and function of nearby genes. Specifically, we expect ultraconserved TAD boundaries to be associated with genes whose disruption interferes with fundamental biological functions. Consistently, using the Genomic Regions Enrichment of Annotations Tool (GREAT[36]) to investigate ontology terms associated with TAD boundaries ("Methods") revealed significant enrichment of genes implicated in 'anatomical structure formation during embryogenesis' nearby ultraconserved boundaries in the human genome (GO: 0048646, *q* < 0.05; Supplementary Data 9). To evaluate the importance of ultraconserved TAD boundaries in proper gene regulation and function, we used CRISPR-Cas9 to

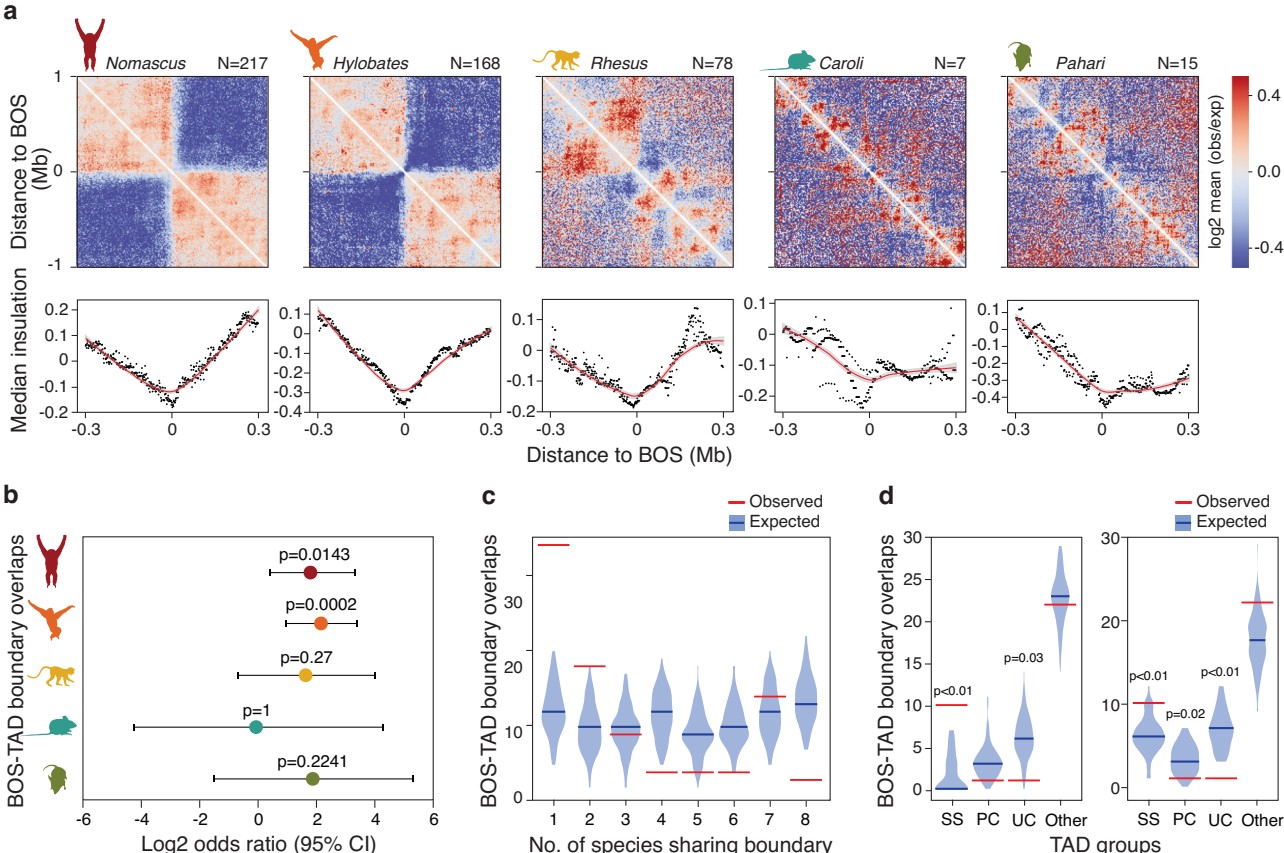

**Fig. 3 | BOS are associated with species-specific TAD boundaries. a** Heatmaps show the frequency of genomic interactions around BOS. Dot plots show median insulation scores, along with loess smoothed curves in red. Error bands represent 95% confidence intervals and are shaded in gray. **b** Log2 odds ratio for the overlap between BOS and TAD boundaries is shown across genomes and error bars represent 95% lower and higher confidence interval. *P*-values based on Fisher's exact test without multiple tests adjustments (Nomascus $n_{BOS}$ = 217, hylobates $n_{BOS}$ = 168, rhesus $n_{BOS}$ = 78, caroli $n_{BOS}$ = 7, pahari $n_{BOS}$ = 15). **c** Violin plots show the expected distribution of cross-species conservation levels of TAD boundaries

overlapping BOS in all species (*n* = 90). Blue lines represent median random expectation, and red lines represent observed values in each category. **d** Observed and expected frequency distribution of evolutionary conservation level of TAD boundaries overlapping BOS in Nomascus (*n* = 34) and Hylobates (*n* = 43) is shown. Blue lines represent the median expected value, and red lines represent the observed value in each category. Empirical *p*-values are calculated based on one-sided permutation tests. SS Species-specific, PC primate-conserved, US ultra-conserved. *P*-values reported only when <0.05 and no adjustments were made for multiple comparisons.

investigate the functionality of a highly conserved TAD boundary in mouse (Fig. 1). This candidate boundary was selected based on the following criteria: (1) not overlapping protein-coding genes; (2) containing cross-species conserved CTCF binding site; and (3) presence of development-associated genes flanking the TAD boundary. To remove our candidate ultraconserved boundary we deleted a ~18Kb region (mm10, chr2:115,840,641-115,858,055) upstream of *Meis2*, a TALE transcription factor involved in inner-ear[37] and heart development[38–40] (Fig. 4a). We obtained four stable mouse lines (*B5234*) that were validated via PCR, sequencing and Southern blot, and used two of them for downstream phenotyping (Supplementary Fig. 8, Supplementary Note 5). Due to the known role of *Meis2* in heart development[38–40], we investigated the consequences of this deletion in this organ first. At the molecular level, CTCF ChIP-seq confirmed successful removal of the targeted CTCF binding sites (Fig. 4a) and qPCR showed significant upregulation of *Meis2* expression in the heart tissue of *B5234*[−/−] mice (Fig. 4b). Comparison of the mutant and wild-type heart using Capture Hi-C[37] revealed chromatin interaction changes consistent with loss of TAD boundary insulation and merging of the neighboring TADs (i.e., increased interactions across deleted boundary and loss of local minima in insulation score; Fig. 4c). At the morphological level, visual comparison of cardiac histology (H&E staining) of *B5234*[+/+] and *B5234*[−/−] mice (*n* = 4/group) showed no apparent overt cardiac defects, including no differences in cardiac wall thickness between the left and right

ventricles (LV and RV, respectively). However, LV walls in *B5234*[−/−] mice appeared more compact with less extracellular tissue/space in between cardiac cells and displayed irregular trabeculation in some regions (Fig. 4d). Comparison of extracellular space within the LV cardiac wall revealed that *B5234*[+/+] mice exhibited a significantly larger percentage of extracellular space than *B5234*[−/−] mice (Student's *t*-test; *p* = 0.00004) (Fig. 4e).

Given *Meis2*'s role in ear development[37], we also characterized the morphology of the utricle and the cochlea in neonatal mice (Supplementary Note 6) and found that these structures remained unchanged among all three genotypes (*B5234*[+/+], *B5234*[+/−], *and B5234*[−/−]). Hair cells appeared to have normal morphology and bundle orientation in both the cochlea and utricle (Supplementary Fig. 9). Consistent with these findings, no differences were found in the hearing levels between the three genotypes measured by auditory brainstem response (ABR) recordings in 7-month-old mice (Supplementary Note 6). Overall, the deletion of ultraconserved boundaries resulted in tissue-specific gene expression changes and mild but significant phenotypic changes relevant to the biological function of the genes near the boundary.

## Deletion of a human-specific boundary alters gene expression in human neurons

The emergence of new TAD boundaries changes local TAD organization, which could lead to the evolution of novel and adaptive gene

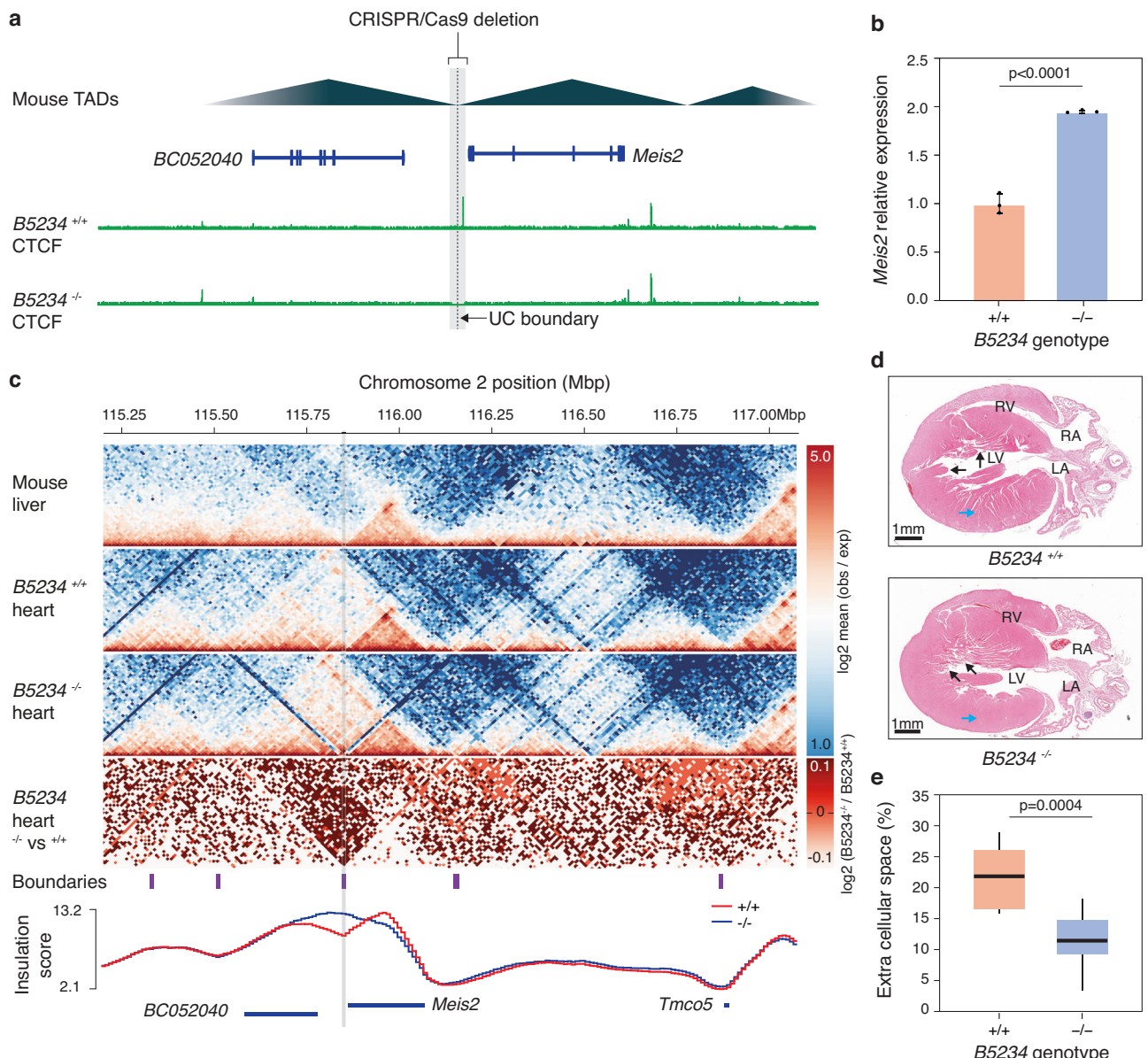

**Fig. 4 | Hearts of *B5234⁻/⁻* mice exhibit changes in *Meis2* gene interactions, expression and histology. a** Schema of the deletion in *B5234⁻/⁻* mice. **b** RT-qPCR reveal increased *Meis2* expression in hearts of *B5234⁻/⁻* mice. Data is presented as mean +/- stdev (two-tailed Student's *t*-test *p*-value < 0.0001; *n* = 3 biological replicates). **c** Top to bottom, heatmaps correspond to Hi-C matrix of mouse liver, Capture Hi-C of *B5234⁺/⁺* and *B5234⁻/⁻* hearts, and log2 fold-change of *B5234⁻/⁻* vs. *B5234⁺/⁺*. Difference in TAD boundary annotations from mouse liver are marked with purple bars below. Deleted ultraconserved boundary is shaded gray. **d** H&E staining of longitudinal sections of *B5234⁻/⁻* and *B5234⁺/⁺* mouse hearts show

differences in LV compaction (blue arrow) and trabecular structures (black arrows). **e** Box and whisker blot based on H&E images confirms increased compaction (decreased extracellular space) in hearts of *B5234⁻/⁻* in comparison to *B5234⁺/⁺* mouse hearts. The ends and center line of the box represent 1st, 2nd (median) and 3rd quartiles. Whiskers extend to minimum and maximum values (two-tailed Student's *t*-test *p*-value = 0.0004; *n* = 4 per genotype). LV Left ventricle, RV Right ventricle, LA Left atrium, RA Right atrium. No *p*-value adjustments were made for multiple comparisons.

regulatory modules. To this end, among the different boundary conservation groups, the human-specific boundaries are of particular significance as they may be relevant to the evolution of human-specific traits and adaptations. When investigating gene ontologies (GO) associated with human-specific TAD boundaries, we detected a significant enrichment of pathways pertaining to 'positive regulation of synapse assembly' (GO:0051965, *q* < 0.05; Supplementary Data 10). Synapse formation is one of the main brain development processes that sets humans apart from other primates[41] and its pathology sits at the heart of several human cognitive disorders, such as autism spectrum disorders (ASD) and schizophrenia[42]. Human-specific boundaries were also associated with several genes implicated in brain disease and

development, such as the Autism Susceptibility Candidate 2 (*AUTS2*) gene, a key regulator of transcriptional networks and a mediator of epigenetic regulation in neurodevelopment. *AUTS2* is implicated in ASD and other neurological diseases[43,44] and contains the most significantly accelerated genomic region differentiating humans from other primates[43]. To further investigate the association of human-specific TAD boundary deletions with human disease, we tested their overlap with pathological copy number variants (CNVs), previously identified from 29,085 children with developmental delay and 19,584 healthy controls[45]. Using permutation analysis (Supplementary Note 3), we found significantly higher recurrency of human-specific TAD boundary deletions in children with developmental delay (*n* = 335

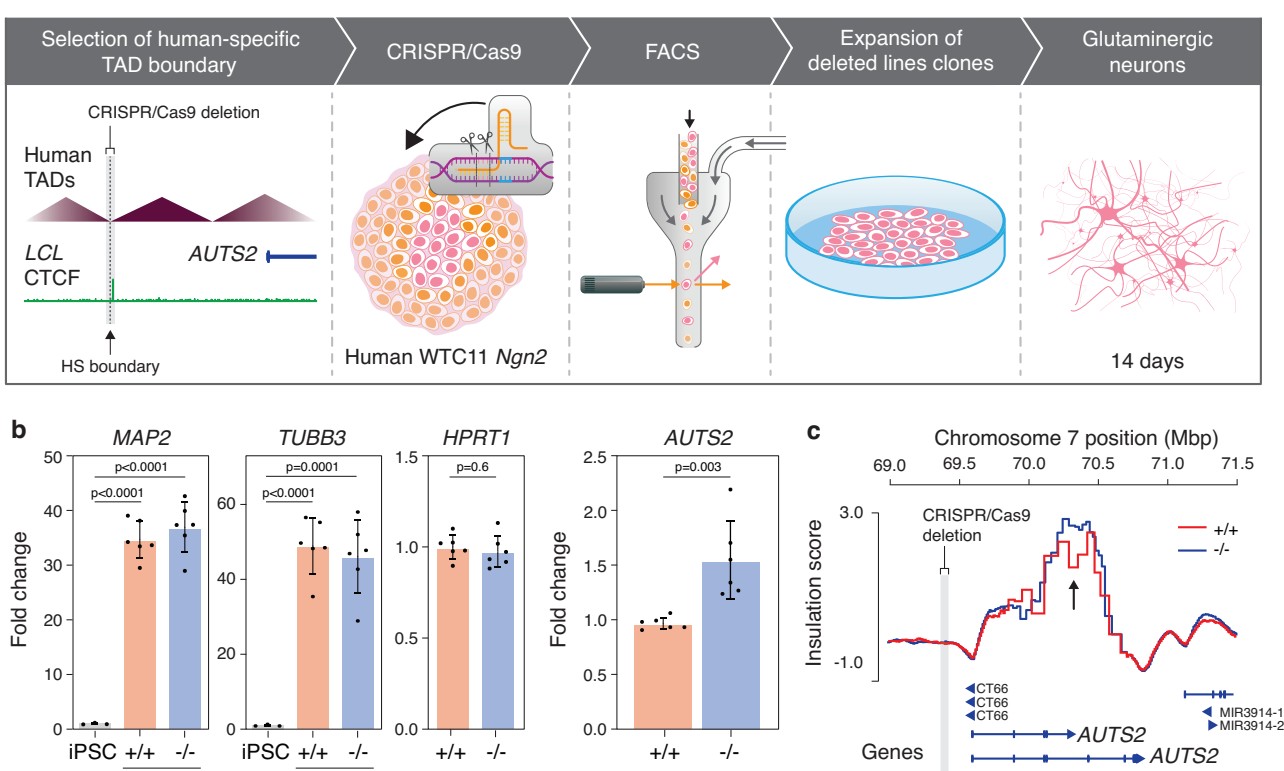

**Fig. 5 | iPSC-differentiated *B14804⁻/⁻* neurons have lower *AUTS2* expression and higher local insulation score. a** Flowchart shows steps involved in generating *B14804⁻/⁻* neurons. **b** Differentiated *B14804* cells show higher expression of neuronal markers, *MAP2* and *TUBB3*. *B14804⁻/⁻* neurons show no change in *HPRT1* expression but have higher *AUTS2* expression based on two-tailed Student's *t*-test.

Data presented as mean ± stdev (*n* = 2 biological replicates, measured over 3 technical replicates). *P*-values were not adjusted for multiple comparisons. **c** A local increase in insulation score (i.e., more interaction passing) is observed at the *AUTS2* locus downstream of the *B14804* deletion.

out of 1130 human-specific boundaries) compared to healthy controls (*n* = 161) (Fisher's exact test, *p* < 0.0001).

We next characterized the function of a human-specific boundary (B14804) in human neurons located ~200Kb upstream of *AUTS2*, by using CRISPR-Cas9 to delete an 11Kb region (hg38: chr7:69,393,633-69,405,124). The deletion was performed in induced pluripotent stem cells (iPSC) WTC11-Ngn2 cell line, which allows quick and robust differentiation to neurons[46] (Fig. 5a). We obtained two lines carrying independent deletions (*B14804⁻/⁻* cell lines 1 and 2) and validated them via PCR (Supplementary Data 11). Both lines, along with the wild-type line, as a negative control, were differentiated into dopaminergic neurons, as confirmed by expression of neuronal marker genes *MAP2* and *TUBB3* (Fig. 5b). We found *AUTS2* expression to be significantly increased in both lines compared to wild-type cells (*p* = 0.0031; Student's *t*-test), while no changes were detected in expression of the *HPRT1* housekeeping gene (Fig. 5b). We also carried out Hi-C on the two *B14804⁻/⁻* and wild-type neuronal cell lines. Despite no visible changes in the interaction matrix (Supplementary Fig. 10), comparing the insulation scores between *B14804⁻/⁻* and *B1480⁺/⁺* neurons revealed notably higher insulation score at the *AUTS2* locus in *B14804⁻/⁻* cells, indicating the deletion results in overall more genomic interactions passing across parts of the *AUTS2* locus (Fig. 5c). Altogether, our findings not only hints at the potential involvement of human-specific TAD reorganization in the evolution of the complex human brain but also suggests that disruption of human-specific TAD boundaries could contribute to gene misregulation and potential developmental and neurological disorders.

## Discussion

In this study, we investigated mammalian TAD conservation by obtaining and comparing high-resolution Hi-C and ChIP-seq data across four primate and four rodent species (Fig. 1). We estimated that the genomic placement of 14% of human and 15% of mouse TAD boundaries was remarkably conserved across all species. These proportions are much lower than those previously reported between human and mouse[7], challenging the long-standing notion that TAD organization is highly conserved across tissues and species. It should be noted however, that we used relatively strict criteria for inferring conservations by requiring boundaries to successfully lift over to the human reference genome and only considered boundaries as overlapping if they were within 10 Kb of each other. Furthermore, our cross-species Hi-C datasets originated from two different cell types, LCL and liver, and thus any boundary identified as ultraconserved had to be developmentally stable between these two tissues, adding constraints to our classification criteria. Nonetheless, our data highlights that boundaries in a genome vary in their level of evolutionary conservation, and this variation is associated with overall differences in their genetic and epigenetic properties. For example, we observed that ultraconserved TAD boundaries have higher insulation strength and harbor more CTCF binding sites compared to species-specific boundaries. Contributing to this pattern may be the fact that boundaries that have lower insulation strength and fewer CTCF binding sites are more likely to be missed during boundary annotation, resulting in arbitrarily lower conservation estimates. Nevertheless, these findings are in line with a previous study that reported higher CTCF binding site

clustering and stronger CTCF affinity at TAD boundaries shared across five murine species, compared to boundaries with lower conservation[47]. Moreover, CTCF binding sites at ultraconserved boundaries overlap with older transposable elements (namely MIRs), whereas human-specific boundaries are associated with more recent families (ERVs), reinforcing the notion that different waves of transposable elements have been shaping genome topology through the insertion of CTCF binding sites at different evolutionary times[25,27,30]. Altogether, this indicates that recurrent emergence and maintenance of several, possibly redundant, CTCF binding sites contribute to the higher conservation and insulation strength of ultraconserved TAD boundaries.

TAD boundaries are shown to contribute to regulating the expression of nearby genes[48], and their disruption can lead to gene misregulation and aberrant phenotypes[4–6,49]. Hence, it has been proposed that TAD boundaries evolve under evolutionary constraints that prevent their disruption even in the face of evolutionary chromosomal rearrangements[12,14]. In support of this hypothesis, previous studies have shown a co-localization of evolutionary breaks of synteny (BOS) and TAD boundaries, even in species with highly rearranged genomes, like gibbons[12,14,50,51]. Here, we also report significant co-localization between BOS and TAD boundaries in the gibbon species, which carry highly rearranged genomes[13]. Additionally, our evolutionary classification enabled us to detect over-representation of species-specific (i.e., more recent) TAD boundaries among those that co-localize with BOS, across all species and within each gibbon genome (Fig. 3c, d). Consistently, we found that the rearrangement events associated with gibbon BOS were also mostly species-specific. All in all, these observations suggest co-occurrence between some evolutionary chromosomal rearrangements and the establishment of TAD structures. Although the underlying mechanism for this co-occurrence is unknown, co-option of existing insulating elements (e.g., TEs containing CTCF binding sites) at the BOS may be one of the first steps toward the establishment of an evolutionarily stable TAD boundary. This hypothesis is indirectly supported by the enrichment of TEs at both BOS and TAD boundaries[35]. It is worth noting that despite the statistically significant over-representation of TAD boundaries near BOS and consistent with previous reports[14], many gibbon BOS do not overlap with TAD boundaries in our dataset, highlighting the complex gene regulatory outcomes that may follow evolutionary genomic rearrangements due to TAD reorganization.

An important implication of the functional connection between TAD boundaries and genes is that selection pressures under which TAD boundaries evolve are likely influenced by the function of nearby genes. Consistently, in the human genome, we observed that genes associated with ultraconserved boundaries were enriched in pathways pertaining to anatomical structure formation during embryogenesis, whose disruption will likely lead to developmental defects. Moreover, manipulating an ultraconserved TAD boundary in a mouse model altered interactions and expression of a neighboring gene and resulted in putatively maladaptive heart phenotypes. While we did not detect acute phenotypes, a recent study[19] performing large (35.3 ± 26.5 Kb; mean ± stdev) in vivo deletions of TAD boundaries near developmental genes in mice observed severe phenotypes when two boundaries classified as ultraconserved in our dataset were deleted. Of note, the severity of phenotypes reported in this study seems to broadly correlate with the deletion size and number of CTCF clusters removed. As our deletion was smaller in size and removed just one CTCF binding site, it is possible that retention of functionally redundant neighboring CTCF binding sites mitigated the effects of the deletion.

We also identified TAD boundaries exclusively present in the human genome and showed that they were significantly associated with genes implicated in synapse assembly. Of note, the human brain undergoes unique synapse patterning, which is thought to contribute to its extraordinary complexity and connectivity[41]. For instance, the timing of synaptic development and expression of relevant genes is prolonged in humans compared to macaque and chimpanzee, and synapse density is highly increased in humans in comparison to other primates[52]. Mounting evidence also places synapse assembly and its pathology at the heart of several human cognitive disorders, such as ASD and schizophrenia[42], suggesting that disruption of human-specific boundaries might also be a contributing factor in human neuro-pathology. It should be noted, however, that this study used Hi-C data from liver and LCL tissues to identify human-specific boundaries, and given that TAD organization can vary across tissues[22], the association of human-specific boundaries with brain-related genes and pathways will likely be stronger if investigated directly in brain cells[22]. Nevertheless, we found significantly more overlap between human-specific TAD boundaries and CNVs identified in children with developmental delay[45], compared to healthy controls. This is in line with recent work that found TAD-shuffling to alter enhancer-promoter interactions and be associated with a variety of human developmental disorders[53]. We additionally showed that deletion of a candidate human-specific TAD boundary upstream of the *AUTS2* gene resulted in differences in genomic interaction patterns (i.e., insulation score) and gene expression at the *AUTS2* locus in iPSC-derived human neurons, providing evidence for the functionality of human-specific boundaries in human brain cells.

In summary, we generated high-resolution Hi-C and ChIP-seq data from several species and used direct multi-species comparison to stratify TAD boundaries in the mouse and human genomes based on evolutionary conservation. Our study highlights the remarkable range of diversity that exists in the evolutionary conservation of TAD boundaries, regardless of species (mouse vs. human) and tissue (LCL vs. liver). Furthermore, by comparing genetic and epigenetic features of boundaries across conservation groups and investigating the function of candidate TAD boundaries via in vivo and in vitro manipulations, we provided new insights into the functional implications of TAD boundary evolution. While our study underlines the importance of TAD conservation in maintaining proper gene regulation patterns during evolution, it also showcases how recent changes in TAD organization can contribute to the emergence of evolutionary novelties and species-specific traits. Future comprehensive comparative studies in additional tissues and developmental stages will help us better understand the contributions of TAD organization to biological function, as well as if and how disrupting these structures can lead to pathology.

## Methods

### General
Sex effects were not considered in the study design.

### Hi-C library generation, TAD identification and evolutionary classification
We used liver tissue and lymphoblastoid cell lines (LCLs) to generate Hi-C libraries with the Arima Hi-C Kit following the manufacturer's protocol. Briefly, ~10 mg frozen pellets of the homogenized fixed liver or LCL were lysed and conditioned before chromatin digestion. The digested chromatin was then filled in and biotinylated before ligation. Next, chromatin was protein-digested and reverse crosslinking overnight, followed by purification. The purified DNA was then sonicated using the bioruptor pico (Diagenode) and size selected before library preparation using the NEB DNA Ultra II, following Arima's protocol. The Hi-C DNA was bound to streptavidin beads before enzymatic end prep, adapter ligation, DNA release by heat incubation, and lastly, PCR to barcode and amplify the libraries. Libraries were sequenced on the Illumina HiSeq2500 or NovaSeq6000. Raw Hi-C sequencing was processed using the HiCUP pipeline[54]. Alignments were converted to a multi-resolution Hi-C matrix using pairtools (https://github.com/mirnylab/pairtools). Hi-C data between biological replicates were

combined after verifying their correlation based on the Pearson correlation as calculated by the HiCExplorer package[55] (Supplementary Fig. 1a). We used HiCRes[56] to estimate the resolution of the Hi-C data for each species (Supplementary Data 1). Based on resolution estimates, TAD boundaries were called at 10Kb resolution using the hicFindTADs command from HiCExplorer using the flags "--minDepth 100000 and –maxDepth 600000" (Supplementary Data 1). Briefly, this program uses running windows of different sizes to measure interaction separation or "insulation score" between the two sides of each Hi-C matrix bin. The "insulation score" quantifies the genomic interactions passing across each genomic bin by calculating the mean $z$-score of contacts using sliding windows. Smaller insulation score values indicate fewer interactions (higher interaction separation). Thus, TAD boundaries are annotated by identifying statistically significant local minima in insulation score curves.

To perform cross-species comparisons and determine TAD boundary evolutionary conservation, we lifted TAD boundaries identified in all species to the hg38 coordinates, generated a union boundary map and used paradigms of boundary presence/absence across species to stratify union boundaries into groups, as detailed in the Supplementary Note 2. Boundaries that failed to lift over from the original genome to hg38 coordinates were not included in the union boundary table.

### Epigenetic and genetic characterization of TAD boundaries across conservation groups

We compared insulation score, chromatin structure, CTCF binding sites, and overlap with genes and transposable elements (TE) across TAD boundary conservation groups in human and mouse genomes. Boundary insulation scores were generated as part of the TAD boundary annotation analysis and compared using the Wilcoxon signed-rank test. CTCF binding sites and chromatin states were determined by using a combination of public and newly generated H3K4me1, H3K27ac, K3K4me3, H3K27me3 and CTCF ChIP-seq data from a female and a male of each of the eight species examined in this study (Supplementary Data 2). All rodent and rhesus ChIP-seq libraries were generated from liver tissue, while lymphoblastoid cell lines (LCL) were used for the rest of the species. ChIP-seq libraries were generated and analyzed as described in Supplementary Note 4. Refseq hg38 and mm10 gene annotations were intersected with TAD boundary coordinates, using bedtools[57], to determine the proportion of boundaries overlapping with genes. We used the TEanalysis tool[58] to examine TE enrichment at CTCF ChIP-seq peaks (i.e., binding sites) within TAD boundaries in each conservation group.

### Investigating overlap of breaks of synteny (BOS) with TAD boundary groups

We identified all breaks of synteny (BOS) of evolutionary chromosomal rearrangements using a custom pipeline in which all pairwise comparisons of known chromosomes of the "target" genome are aligned to the "query" genome using LASTZ (https://github.com/carbonelab/lastz-pipeline), followed by alignment chaining and filtering using UCSC tools[56]. For Nomascus, we took advantage of an improved version of the genome assembly (Asia_NLE_v1) based on PacBio CLR and guided by Nleu3 generated by the Eichler lab (Supplementary Note 1). Pairwise genome alignments were then processed using a custom Python script (https://github.com/carbonelab/axtToSyn), which elongates alignment blocks that are longer than 1Kb and have a minimum alignment score of 100,000 by merging them with other blocks on the same chromosome and strand. Elongated alignment blocks represent synteny blocks between two genomes, thus synteny breakpoints were defined as 1Kb regions flanking each elongated synteny block in the target genome. To transfer coordinates of synteny breakpoints from the target genome coordinates to those of the query genome, we used BLAT[59] with the following parameters: -stepSize=5 -repMatch=2253

-minScore=20 -minIdentity=0. The BLAT results were manually evaluated and if the BLAT score of the second highest-scoring hit was within 10% of the top-scoring hit, the breakpoint was annotated as duplicated and removed. Breakpoints that survived this filtration step were further manually inspected to remove BOS overlapping segmental duplications and large repeats. Only curated breakpoints that survived both filtration steps were used for downstream analysis (Supplementary Data 6).

Using custom scripts (https://github.com/carbonelab/hicpileup) we visualized aggregate Hi-C contact frequencies in a 2 Mb window centered at synteny breakpoints. The median insulation score was visualized in 600 Mb windows centered at synteny breakpoints. Since BOS are often found in GC-rich regions, we used stratified random sampling (based on %GC content) to generate a random set of BOS with similar GC content. We used Fisher's exact test to compare the observed number of TAD boundary overlapping BOS (at least 1 bp) (Supplementary Data 7 and 8) to the expected numbers based on the random BOS. We next identified all TAD boundaries located within 30Kb of a BOS in the genomes of two gibbons that had significant overlap between BOS and TAD boundaries. In each species, we matched each overlapping boundary to its corresponding union boundary. To avoid underestimating the evolutionary conservation of boundaries due to lack of synteny, boundaries that did not lift over to the hg38 coordinates were not included in downstream analysis. We then used the presence/absence of each boundary in other species to categorize it as "species-specific", "primate-conserved", "ultraconserved" or "other". For each species, the number of boundaries categorized in each of these groups was compared to expectation, which was determined based on randomly selecting the same number of TAD boundaries from the whole genome. We repeated this process 100 times, and in each boundary group, the proportion of times out of 100 when the observed number of TADs was more extreme than random expectation was the empirical one-tailed $p$-value (Fig. 3c, d).

### Generation of knockout mice

All transgenic animal experiments were conducted on FVB/NJ (Jackson Lab; 001800) mice in accordance with the Guide for the Care and Use of Laboratory Animals established by the National Institutes of Health. Protocols were approved by the Institutional Animal Care and Use Committees (IACUC) at UCSF. All mice were allowed ad libitum access to food and water and were maintained on a 12 h light/dark cycle in a climate-controlled facility at 40–60% humidity and at room temperature. Our CRISPR deletion ($B5234^{-/-}$) targeted an 18Kb region (chr2: 115,840,641-115,858,055, mm10) including an ultraconserved boundary near the *Meis2* gene using i-GONAD[60] described in Supplementary Note 5. Offspring were screened for deletion using Southern blot, sequencing and a custom-designed PCR assay, with primers flanking the deletion site (Supplementary Data 11). Molecular phenotyping and histology assays are described in detail in the Supplementary Note 6.

### Generation and phenotyping of the $B14804^{-/-}$ knockout cell line

We performed CRISPR knockout assays targeting an 11Kb region (hg38: chr7:69,393,633-69,405,124) that included the human-specific B14804 boundary in WTC11-ngn2 cells[46] (i.e., WTC11 cells with a doxycycline-inducible mouse Ngn2 transgene). Briefly, WTC11-ngn2 cells were cultured in mTeSR1 media (STEMCELL Technologies) with daily media changes following normal WTC11 maintenance protocols. Cells were seeded at a density of 300k cells per 6-well in mTeSR1 media plus Rock Inhibitor (Selleckchem) and cultured for one day. WTC11-ngn2 p37 ± 21 cells (p37= passage number before ngn2 introduction, +21= passage number after the ngn2 insertion) were then transfected with 800 ng of each of the four sgRNAs (Supplementary Data 11), 6250 ng of TrueCut Cas9 Protein v2 (Invitrogen), and 500 ng of MSCV Puro-SV40:GFP plasmid (Addgene) using Lipofectamine CRISPRMAX Cas9 Transfection Reagent (Thermo Scientific) following the

manufacturer's protocol. On the second day post-transfection, cells were washed in 1X PBS, dissociated from the plate using Accutase (STEMCELL Technologies), quenched with 1X PBS, spun down and resuspended in a FACs buffer consisting of 1X PBS, 0.5 M EDTA (Neta Scientific), 1 M HEPES PH7.0 (Neta Scientific), 1% FBS, and Rock Inhibitor. Cells were filtered through a cell strainer, then GFP-positive single cells were sorted on a BD FACSAria Flow Cytometer or equivalent using a 100-micron nozzle into 96-well plates containing mTeSR media supplemented with Rock Inhibitor, 1% Penicillin-Streptomycin (ThermoFisher), and 10% CloneR2 (STEMCELL Technologies). Individual colonies were expanded incrementally when wells became confluent. DNA was extracted from a subset of cells of each colony using AllPrep DNA/RNA Mini kit (Qiagen). To validate the deletions, gDNA was extracted using AllPrep DNA/RNA Mini kit (Qiagen), followed by genotyping of each colony using KOD One PCR Master Mix (DiagnoCine) with two unique primer sets (Supplementary Data 11). Passage 37 ± 27 WTC11-ngn2 cell lines were differentiated into day 14 neurons following a previously described protocol. In short, cells were seeded and grown in pre-differentiation media. On the third day, cells were dissociated, counted, and plated in differentiation media according to recommended seeding densities. Cells were grown for 14 days, with only a partial media change on day 7. On day 14, DNA and RNA were extracted from the $B14804^{-/-}$ and wild-type cells using AllPrep DNA/RNA Mini kit (Qiagen). cDNA was synthesized from extracted RNA using SuperScript™ III Reverse Transcriptase (Invitrogen) following the manufacturer's protocol. cDNA was diluted 1:10 and used for qPCR with SsoFast EvaGreen Supermix (BioRad). qPCR reactions were done in triplicate and normalized against *Gapdh*. Frozen $B14804^{-/-}$ and $B14804^{+/+}$ neurons were used to generate Hi-C libraries using the Arima Hi-C Kit as described earlier ($n = 2$ per genotype). Hi-C libraries were sequenced on the Illumina NovaSeq6000 platform. Raw Hi-C data was merged across replicates and interaction matrices and insulation scores were calculated as described above (Supplementary Fig. 10).

### Reporting summary

Further information on research design is available in the Nature Portfolio Reporting Summary linked to this article.

## Data availability

All Hi-C, ChIP-seq and Capture Hi-C data generated in this study are available under the accession number: GSE197926. The reference genomes used in this study are: Human, hg38 [https://genome.ucsc.edu/cgi-bin/hgGateway?hgsid=1738767972_ryzhdYsyt1Nt9sG6pkyJwcl4cRAA]; *Nomascus leucogenys*, Asia_NLE_v1 [https://www.ncbi.nlm.nih.gov/datahub/genome/GCF_006542625.1/]; Hylobates moloch, HMol_V3 [https://www.ncbi.nlm.nih.gov/datasets/genome/?taxon=81572]; Rhesus macaque, rheMac10 [https://genome.ucsc.edu/cgi-bin/hgGateway?hgsid=1738767972_ryzhdYsyt1Nt9sG6pkyJwcl4cRAA]; *Mus musculus*, mm10 [https://genome.ucsc.edu/cgi-bin/hgGateway?hgsid=1738767972_ryzhdYsyt1Nt9sG6pkyJwcl4cRAA]; *Mus pahari*, PAHARI_EIJ_v1.1 [https://www.ncbi.nlm.nih.gov/datasets/genome/GCF_900095145.1/]; *Mus caroli*, CAROLI_EIJ_v1.1 [https://www.ncbi.nlm.nih.gov/datasets/genome/GCF_900094665.1/]; *Rattus norvegicus*, rn6 [https://genome.ucsc.edu/cgi-bin/hgGateway?hgsid=1738767972_ryzhdYsyt1Nt9sG6pkyJwcl4cRAA]. Public Hi-C datasets used in this study were obtained from GEO under the following accession numbers: GSE128800, SRR6675327, GSM3682186 and GSM3682187. Public ChIP-seq data was accessed at GEO under the following accession numbers GSE50893, GSE136963, GSE136968, GSE60269, GSM1087083 and at the EMBL-EBI under https://www.ebi.ac.uk/biostudies/arrayexpress/studies/E-MTAB-1511/sdrf. Source data are provided with this paper.

## Code availability

The following custom scripts were also used and are available on GitHub: LASTZ [https://github.com/carbonelab/lastz-pipeline],

Pairwise genome alignments: https://github.com/carbonelab/axtToSyn, Snakemake workflow for QC and processing of Hi-C data: https://github.com/carbonelab/hic_workflow.

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

## Acknowledgements

The ChIP-seq assays were performed by the KCVI Epigenetics Consortium at OHSU. Next-generation libraries were sequenced at the OHSU Massively Parallel Sequencing Shared Resource, the Genomics and Cell Characterization Core Facility at the University of Oregon and Novogen Co. Data analyses were performed on the Exacloud computer cluster at OHSU. The ONPRC Integrated Pathology Core provided support services for the research. WTC11-ngn2 cells were a kind gift from Li Gan (Gladstone Institute). We thank Drs. Kent Thornburg and Alina Maloyan for reviewing heart images. We additionally thank Drs. Fudenberg and Pollard for sharing the CNV calls from patients and controls. This work was supported in part by National Human Genome Research Institute (NHGRI) Grant R01HG010333 (for L.C. and N.A.), and L.C. and D.F.C. are supported by the National Institute of Health Office of Directors (NIH/OD) Grant P51 OD011092 (to the Oregon National Primate Research Center). Part of this work was supported by the OHSU University Shared Resources shared grant OHSU-USR-2018 to L.C. B.N. is supported by the S10OD021717-01A1 Grant (PI: M. Calvert). E.E.E. is supported by National Institutes of Health Grant R01HG002385. E.E.E. is an investigator of the Howard Hughes Medical Institute.

## Author contributions

L.C., M.O. and N.A. designed the study. K.A.N., L.H., W.L., C.E.L., S.W., J.H., J.W., R.R.S., Y.M., B.N., A.C.L., A.M.S., R.Y., L.F., I.R.M., S.A.E., D.K.C., T.A.J., E.E.E., S.R. and D.F.C. carried out experiments and contributed to data collection. W.L. generated and managed the transgenic mice. J.V. and M.O. performed next-generation sequencing data analysis and J.V. developed all custom scripts. M.O. and K.A.V.C. designed and generated figures. M.O., J.V., N.A. and L.C. wrote the paper. All co-authors provided feedback on the manuscript.

## Competing interests

E.E.E. is a scientific advisory board (SAB) member of Variant Bio, Inc. N.A. is the cofounder and on the scientific advisory board of Regel Therapeutics and receives funding from BioMarin Pharmaceutical Incorporated. The remaining authors declare no competing interests.

## Additional information

[1]Department of Medicine, Knight Cardiovascular Institute, Oregon Health and Science University, Portland, OR, USA. [2]Department of Bioengineering and Therapeutic Sciences, University of California San Francisco, San Francisco, CA, USA. [3]Institute for Human Genetics, University of California San Francisco, San Francisco, CA, USA. [4]Department of Genome Sciences, University of Washington School of Medicine, Seattle, WA, USA. [5]Histology and Light Microscopy Core Facility, Gladstone Institutes, San Francisco, CA, USA. [6]Division of Genetics, Oregon National Primate Research Center, Beaverton, OR, USA. [7]OHSU Transgenic Mouse Models Core Lab, Oregon Health and Science University, Portland, OR, USA. [8]Department of Otolaryngology-Head and Neck Surgery, University of California, San Francisco, CA, USA. [9]Department of Otolaryngology-Head and Neck Surgery, Vanderbilt University Medical Center, Nashville, TN, USA. [10]Howard Hughes Medical Institute, University of Washington, Seattle, WA 98195, USA. [11]Department of Biomedical Engineering, Oregon Health and Science University, Portland, OR, USA. [12]Department of Molecular and Medical Genetics, Oregon Health and Science University, Portland, OR, USA. [13]Department of Medical Informatics and Clinical Epidemiology, Oregon Health and Science University, Portland, OR, USA. [14]Present address: Bio-X Institutes, Key Laboratory for the Genetics of Developmental and Neuropsychiatric Disorders, Ministry of Education, Shanghai Jiao Tong University, Shanghai, China. [15]These authors contributed equally: Mariam Okhovat, Jake VanCampen. ✉e-mail: okhovat@ohsu.edu; nadav.ahituv@ucsf.edu; carbone@ohsu.edu

