## [Peer Review File · Nature Communications]

TAD Evolutionary and functional characterization reveals diversity in mammalian TAD boundary properties and functionEditorial Note: Parts of this Peer Review File have been redacted as indicated to remove third-party material where no permission to publish could be obtained.

REVIEWER COMMENTS

Reviewer #1 (Remarks to the Author):

This manuscript by Okhovat, VanCampen and colleagues reports a comparative analysis of TAD domain boundaries across 8 mammalian species, in order to investigate how evolution of the 3D chromosomal structure may contribute to phenotypic change in mammals. The authors find that most TAD boundaries are not conserved between mammals or even across closer clades, and frequently overlap recent transposable elements that carry CTCF binding sites. These findings contrast with earliest reports investigating this question at a lower experimental resolution, which reported high conservation of 3D genome structures in mammals, but are consistent with more recent work at high resolution and also other lines of evidence suggesting that 3D genome organization evolves largely under drift in mammals, partly driven by TE expansions. Further, the authors identify hundreds of conserved boundaries across these 8 mammals, which may be maintained due to selection. The authors use genome edition to show that removing 2 of topological boundaries results in nearby gene expression changes and phenotypic modifications (the authors report 3 editions, but one of these does not in fact correspond to a TAD boundary – see comments below).

Altogether, this is an interesting and in-depth investigation of the evolution of genome structure across a relatively large panel of mammals, which will appeal to a wide readership. The results are not particularly groundbreaking, as they largely align with previous findings - e.g. the low conservation of TAD boundaries was previously reported between human and chimp, the contribution of TEs to CTCF binding remodeling has been reported extensively, etc – but as they unify these findings into a consistent, wider evolutionary framework, this should not necessarily impede publication in my opinion. The manuscript however suffers from a number of design and analysis issues that need to be clarified, as they may affect the results quite substantially and perhaps invalidate some of the more novel findings (see comments below).

1. The authors produced Hi-C maps from liver tissue for 5 species, and from LCL cell lines in 3 others (human and two gibbons). As the authors themselves acknowledge in the discussion, TAD structure can vary quite substantially between tissues. How does this affect the analysis of interspecies boundary conservation? Are there boundaries that are found conserved only between the liver or LCL samples? While I understand that this was driven by tissue availability issues, a common issue in primates, it is also a major design flaw to the study and should be addressed accordingly.

2. Quality control and reproducibility information is glossed over in the main text, but should be presented and discussed more extensively as they are crucial to interpret the results. Based on the supplementary information provided, there are substantial differences in usable Hi-C sequencing depth between species, with an almost 2-fold difference in coverage between the top and bottom library (306 and 161 million read pairs, respectively). Sequencing depth impacts statistical power to detect TAD boundaries and some of these libraries are somewhat under-sequenced to call boundaries at a 10kb resolution, as performed here, which typically requires ~300-400M valid read pairs. A substantial fraction (30-40%) of sequencing reads do not align to the reference genome in most libraries, and correlation between replicates is fairly low for some species including humans (see Supp Fig 1). Altogether, these statistics suggest that some boundaries may be missed due to insufficient coverage and/or reproducibility, and that this may affect the 8 species unequally. This needs to be addressed as well, to confirm that the low conservation reported here is not partly driven by false negatives.

3. The detection of conserved boundaries relies on whole-genome alignments and lift-over, which decreases in efficiency with evolutionary distance. Indeed, as reported in the supplements, ~25% of rat TAD boundaries could not be aligned to the human genome. It is unclear how such non-alignable TAD boundaries were handled by the authors. Are these considered as non-conserved? This would be problematic at least for the comparison between TAD boundary conservation and synteny breaks, as whole-genome aligners rely on synteny conservation to chain aligned sequence blocks and therefore frequently fall apart around synteny breaks (e.g. a TAD boundary at a synteny break would very frequently be called as species-specific, due to missing information in the alignment).

4. The synteny break analysis is not clearly explained and difficult to follow. How is a synteny break defined here? Are breaks in non-human primates all defined relative to human? If so, how many rearrangements identified in the two gibbon species are shared between both species – can these analyses truly be considered independent? And how did the authors differentiate breaks that occurred in the human lineage from those in the target species?

5. The authors report significant overlaps between TAD boundaries, TEs (especially SINEs and LTRs), and synteny breaks. Like TAD boundaries, these classes of TEs as well as chromosomal rearrangements are well documented to preferentially occur in gene-rich, GC-rich regions of the genome. It is unclear that the whole genome is an appropriate background here to test co-occurrence and conclude that a plausible causal relationship exists between these genomic structures: at least, the permutation should be adjusted for %GC.

6. I am confused by the first reported example of TAD boundary deletion, where the authors edited a 23 kb region near *Dmrtb1* in a mouse mutant. According to Fig 4a, that deletion does not cover the conserved TAD boundary, but actually targets a nearby CTCF site. I could not find evidence in the manuscript that any disruption of the conserved TAD boundary exists in this mutant – in fact, the authors report no differences in interaction frequencies at that border. The way the authors present this as evidence of the functional importance of ultra-conserved boundaries is highly misleading - it is more likely that the reported gene expression and phenotypic changes are due to the deletion of the CTCF site, as CTCF has many other gene regulatory roles besides its insulating properties.

7. I found the discussion a bit confusing, as the authors strongly emphasize that TAD boundaries likely “evolve under strong evolutionary constraints preventing their disruption”, which seems in direct contradiction with their findings that TAD boundaries are not very conserved and frequently remodeled by TEs.

Minor points

- Figure 1 and 2a-d are largely redundant information. I would suggest to condense.
- “Human LCL genome”, “mouse liver genome”: this is weird phrasing, consider modifying to “human genome in LCL”/“mouse genome in liver”, etc.
- “Deletion of ultraconserved TAD boundaries results in selective phenotypes”: that paragraph title is misleading, the authors do not test for selection here, only functional impact (functional changes can be non-selective)

Reviewer #3 (Remarks to the Author):

Review of the manuscript:

“TAD evolutionary and functional characterization reveals diversity in mammalian TAD boundary properties and function”

How chromatin organization evolves at the level of TADs and its implication in the evolution of gene regulation is a highly interesting and debated topic. The manuscript of Okhovat et al. helps to shed light on this question contributing valuable Hi-C and epigenetic datasets in different species and the analysis of functional experiments in disparate organs like testis and heart. I find the manuscript to be valuable for the community and the readership of Nature Communications. However, I have concerns and I disagree with some of the conclusions presented.

Major points

1. The authors claim that their TAD boundary conservation data challenge the assumption that TAD organization is conserved across tissues / evolution because of the low percentage (14-15%) of boundaries common in the eight different species. I would not interpret the data in that way,

since also more than 50% of the TAD boundaries are conserved in at least 4 other species. It is expected that increasing the number of species used in the comparison also increases the chance that a particular boundary is not found in at least one of the other species. Besides, small differences in insulation can cause boundary calling algorithms to fluctuate specially in the “weak” boundaries regime. Furthermore, the conservation of TAD boundary locations is not necessarily the best way to address conservation of TAD structures (albeit it is perhaps the most straightforward). For instance, a boundary might be displaced 100 kb upstream or downstream in a particular species and count as not conserved. However, the content of the rest of the TAD (which might consist of 500 or 600 kb) in terms of regulatory regions and syntenic genes might be almost identical. If the attention is only focused on the boundary location, this TAD would be considered as entirely divergent even though 90% of its content is conserved. In summary, I believe that only focusing on the fact that “only” 14% of boundaries are conserved in 8 out of 8 species gives the false impression that 80% of the TAD structures in mammals are evolving almost neutrally. Which I think is a notion that the very same data presented in this study speak against.

2. I am concerned that weaker boundaries might be more difficult to find by the boundary caller in other species and this analysis might contain false negatives. This will influence the conclusion that weaker boundaries are less conserved, maybe they are just closer to the limit of being called a boundary or not.

3. The authors claim that BOS-overlapping TAD boundaries were formed later or coordinated with the break of syteny event because they tend to occur at “young” boundaries. I think this is a very intriguing finding that should be discussed further: it challenges the interpretation that BOS happen at boundaries because then TAD structure is preserved. What force would create a boundary after a rearrangement?

4. In the deletion of the CTCF site overlapping the gibbon BOS at the *Dmrt1b* locus, the nature of this BOS is not clarified. Is this an inversion, a translocation, an insertion? Where is the other breakpoint? Is the other breakpoint inside the same TAD or is the BOS altering the TAD structure? It is difficult to interpret the rearrangement effects without this information.

5. The resolution of the Capture-C data from the *Dmrt1b* locus is insufficient to rule out whether there are effects at the level of chromatin organization. I think that at least some 4C-seq (ideally UMI) with sufficient resolution should be provided to clarify the effects of the mutation on the regulatory landscape of *Dmrt1b*. Also, the exact location of the boundary is unclear from the Capture-C data, and the Hi-C data is also not shown. The data suggest enhancer adoptions of *Lrp8* enhancers by *Dmrt1b* but with the current experiments is difficult to tell.

6. Chromosome conformation capture data (either Hi-C, Capture-C or 4C-seq) should be provided also for mutants and controls in the *Meis2* and *AUTS2* locus. In the case of *AUTS2* Hi-Cs are available but only the insulation score is shown.

Minor points

1. I find redundant the content of Fig. 2a-d and Fig. 1.

2. Fig. 2c do not seem to correspond to the statement “species-specific TAD boundaries in human and mouse show higher enrichment of chromatin states associated with active transcription start sites, bivalent chromatin and CTCF signal”. In the heatmaps it looks like the other way around.

3. The labels asterisks and scRNA, tRNA labels of figure 2e are confusing and their meaning is not defined in the caption.

REVIEWER COMMENTS

Reviewer #1 (Remarks to the Author):

This manuscript by Okhovat, VanCampen and colleagues reports a comparative analysis of TAD domain boundaries across 8 mammalian species, in order to investigate how evolution of the 3D chromosomal structure may contribute to phenotypic change in mammals. The authors find that most TAD boundaries are not conserved between mammals or even across closer clades, and frequently overlap recent transposable elements that carry CTCF binding sites. These findings contrast with earliest reports investigating this question at a lower experimental resolution, which reported high conservation of 3D genome structures in mammals, but are consistent with more recent work at high resolution and also other lines of evidence suggesting that 3D genome organization evolves largely under drift in mammals, partly driven by TE expansions. Further, the authors identify hundreds of conserved boundaries across these 8 mammals, which may be maintained due to selection. The authors use genome edition to show that removing 2 of topological boundaries results in nearby gene expression changes and phenotypic modifications (the authors report 3 editions, but one of these does not in fact correspond to a TAD boundary – see comments below).

Altogether, this is an interesting and in-depth investigation of the evolution of genome structure across a relatively large panel of mammals, which will appeal to a wide readership. The results are not particularly groundbreaking, as they largely align with previous findings - e.g. the low conservation of TAD boundaries was previously reported between human and chimp, the contribution of TEs to CTCF binding remodeling has been reported extensively, etc – but as they unify these findings into a consistent, wider evolutionary framework, this should not necessarily impede publication in my opinion. The manuscript however suffers from a number of design and analysis issues that need to be clarified, as they may affect the results quite substantially and perhaps invalidate some of the more novel findings (see comments below).

We thank the reviewer for deeming our work to be of interest and appeal to a wide-readership. We are also grateful for their in-depth evaluation of our study and for providing constructive feedback that has helped us improve our manuscript. Below, we provide point-by-point responses to their concerns. We hope they find our responses and revisions satisfactory.

1. The authors produced Hi-C maps from liver tissue for 5 species, and from LCL cell lines in 3 others (human and two gibbons). As the authors themselves acknowledge in the discussion, TAD structure can vary quite substantially between tissues. How does this affect the analysis of interspecies boundary conservation? Are there boundaries that are found conserved only between the liver or LCL samples? While I understand that this was driven by tissue availability issues, a common issue in primates, it is also a major design flaw to the study and should be addressed accordingly.

As the reviewer points out, we had to use a combination of liver and LCL due to primate tissue unavailability. To improve transparency and allow easier investigation of the patterns of cross-species boundary conservation, we have added a new supplemental table (SupplementaryTable 5) which contains the frequency distribution of boundaries across all possible combinations of cross-species/tissue conservation patterns. Based on this table, of the total 17,930 union boundaries, 273 boundaries are exclusively present in all liver samples (i.e. 5-way conserved boundaries that are only found in rhesus, mouse, rat, caroli and pahari), and 78 that are exclusively found in all LCL samples (i.e. 3-way conserved boundaries that are shared only among human, Nomascus and Hylobates). We have also generated Upset plots to visualize the pattern of cross-species boundary overlap within each level of conservation (1-way, 2-way, ..., 8-way). These plots are now presented as Supplementary Fig. 1.

Unfortunately, it is difficult to fully tease apart the effects of evolutionary relatedness vs. tissue similarity in our data, since all rodent libraries are made from liver and all primates, except rhesus, are from LCL (species and tissues are almost co-linear variables). However, we do observe that rhesus boundaries (liver tissue) coincide more often with boundaries in other primates (phylogenetically related, but different tissues), than with rodents (same tissue, but phylogenetically distant). This suggests that evolutionary distance has a stronger impact on differentiating boundaries across our samples. For the reviewer's consideration, below we present another Upset plot which summarizes the top 40 most prevalent combinations of cross-species union boundary overlaps. This plot shows that following species-specific and ultra-conserved boundaries, primate-conserved (boundaries found in both liver and LCL tissues) are the most common pattern of boundary conservation.

Upset plot summarizing the top 40 most prevalent combinations of cross-species boundary conservation

Consistently, logistic PCA analysis of TAD boundary presence/absence patterns across samples (please see below) reveals grouping of samples based on phylogenetic order (primate vs. rodent) along the first principal component (PC1), indicating that phylogenetic relations explain most of the variation in TAD boundary organization (see Fig below). This figure is now presented in the supplemental materials as **Supplementary Fig. 2**.

[REDACTED]

PCA analysis of TAD boundary presence/absence patterns across samples shows that samples group based on phylogenetic order (primate vs. rodent), suggesting that phylogenetic relations is the strongest variable in explaining variation in TAD boundary organization.

crucial to interpret the results. Based on the supplementary information provided, there are substantial differences in usable Hi-C sequencing depth between species, with an almost 2-fold difference in coverage between the top and bottom library (306 and 161 million read pairs, respectively). Sequencing depth impacts statistical power to detect TAD boundaries and some of these libraries are somewhat under-sequenced to call boundaries at a 10kb resolution, as performed here, which typically requires ~300-400M valid read pairs. A substantial fraction (30-40%) of sequencing reads do not align to the reference genome in most libraries, and correlation between replicates is fairly low for some species including humans (see Supp Fig 1). Altogether, these statistics suggest that some boundaries may be missed due to insufficient coverage and/or reproducibility, and that this may affect the 8 species unequally. This needs to be addressed as well, to confirm that the low conservation reported here is not partly driven by false negatives.

We agree with the reviewer regarding the importance of reporting data quality and coverage, and that is why we made sure to report such statistics in Supplementary Table 1. We would like to clarify that the relatively low and variable library coverages pointed out by the reviewer (ranging from 161 to 306 million read pairs) refer to individual replicates, and as outlined in the manuscript, replicate libraries were merged before calling TAD boundaries, resulting in higher coverages and lower cross-species variability (total valid reads ranging from 338M to 583M across species). We should also clarify that the alignment rates we report in the supplemental table are the proportion of reads that are aligning in proper pairs, thus a subset of the total reads aligning to each genome. We have now corrected the labels of Supplementary Table 1 to clarify this. In fact, the total percent reads aligning to each genome is much higher, ranging from 82% to 96% across samples, and the proportion of reads aligning uniquely ranges from 68% to 82%.

The reviewer was also concerned that we may not have sufficient coverage to call TAD boundaries at a 10Kb resolution. To address this, we used the HiCRes program (Marchal et al. 2022), to estimate the resolution of our Hi-C libraries. This software uses the Rao *et al.* definition of Hi-C resolution, where resolution is defined as the minimum size window which, when used to calculate the genome coverage, leads to 80% of the windows being covered by >1,000 reads. In other words, the Hi-C library resolution is the window size for which 20th percentile of the reads per window equals 1,000. Based on this, we found the resolutions of our Hi-C datasets to range from 5.1-9.9Kb across species, confirming that we do have sufficient coverage to call TADs at 10Kb resolution. Furthermore, we did not

All this said, we agree that the use of two tissues is a caveat of our study, and while we do not expect this issue to change the overall findings of our study, it likely leads to some inaccuracies in the evolutionary classifications of boundaries and may also result in lower estimates of cross-species conservation. Therefore, in addition to the new Supplementary Figures and Tables, we have added some new text discussing the potential impacts of using two tissues (see Lines 147-156 in the main manuscript and Lines 73-83 in the Supplementary notes).

2. Quality control and reproducibility information is glossed over in the main text, but should be presented and discussed more extensively as they are

find any significant correlation (p value=0.8) between a species' overall number of valid pairs vs. number of predicted TAD boundaries in our study, suggesting that sequencing coverage was not a strong bottleneck in our boundary predictions across species. In the revised manuscript, we have added resolution estimates to Supplementary Table 1 and provided more details about the data quality statistics in the Results (see lines 120-124) and Methods (see lines 512-513).

3. The detection of conserved boundaries relies on whole-genome alignments and lift-over, which decreases in efficiency with evolutionary distance. Indeed, as reported in the supplements, ~25% of rat TAD boundaries could not be aligned to the human genome. It is unclear how such non-alignable TAD boundaries were handled by the authors. Are these considered as non-conserved? This would be problematic at least for the comparison between TAD boundary conservation and synteny breaks, as whole-genome aligners rely on synteny conservation to chain aligned sequence blocks and therefore frequently fall apart around synteny breaks (e.g. a TAD boundary at a synteny break would very frequently be called as species-specific, due to missing information in the alignment).

It is true that our rodent boundaries (particularly rats) have lower lift-over success (76-84% across rodents) compared to primates (91-98%). Our approach relies on the successful lift over of boundaries to the human reference genome and boundaries that fail to liftover from their original species genome to human were not included in the union map. As a result, variation in the success of lift-over of a boundary across species could be misinterpreted as non-conservation. We now clarify this in the revised Results and Methods (see lines 261-264, 524-525, and lines 572-573). Furthermore, we have revised the language throughout the manuscript to clarify that our criteria for determining boundary conservation is strict and thus, may underestimate conservation. We have also lessened the emphasis we previously placed on reporting that cross-species boundary conservation is low.

The reviewer is correct that regions of break of synteny could result in arbitrarily low conservation estimates due to inefficient lift-over. We did consider this issue and therefore restricted our evolutionary classification only to those boundaries that were present in the union map and could successfully be lifted over. While this reduced the total number of boundaries we could study by ~40% (we retained 90 out of total 149 TAD boundaries that were located near BOS), it decreases the chances of mischaracterizing conservation levels due to loss of synteny and failed lift-over. We now try to better clarify and justify our approach and specify that we were not able to determine evolutionary conservation of a subset of TAD boundaries overlapping BOS, due to failure to lift over (see lines 260-264)

4. The synteny break analysis is not clearly explained and difficult to follow. How is a synteny break defined here? Are breaks in non-human primates all defined relative to human? If so, how many rearrangements identified in the two gibbon species are shared between both species – can these analyses truly be considered independent? And how did the authors differentiate breaks that occurred in the human lineage from those in the target species?

Breaks of synteny (BOS or synteny-breaks) in primate and rodent genomes were identified through pair-wise alignments against the human and mouse reference genomes, respectively, as done in the past (Carbone et al. 2009, Carbone et al. 2014, and Lazar et al. 2018). This is mentioned in the Methods (lines 542-560), and we have now added some details in the Results (see lines 242-245) and as well as **Supplementary Table 6** and **Supplementary Table 7**. We also used previous curated dataset of *Nomascus*-human BOS (Lazar et al. 2018) to validate our BOS discovery method. Moreover, all BOS were manually curated based on cytogenetics data obtained in the past (Carbone et al. 2006, 2009 and 2014) and, in the case of Pahari and Caroli, published data (Thybert et al. 2018).

We know that the vast majority gibbon-human synteny breaks occurred in the gibbon lineage, and indeed the organization of the human chromosomes are highly similar to that of other apes (i.e. likely ancestral). We also used a triangulation comparison among human, gibbon and rhesus to identify and discard any BOS that may be human-specific, as done in the past (Carbone et al. 2006). The

reviewer is right that some of the breaks of synteny might be shared between the two gibbons. Indeed, from previous work (Capozzi, Carbone et al. 2012 and Lazar et al. 2018) we know that roughly 50% of all the rearrangements present in the gibbon lineage occurred in the gibbon ancestor before the split of the four gibbon genera ~5 million years ago, and are therefore shared among all gibbons. However, the majority of BOS that overlap TAD boundaries in each species (86% in *Nomascus* and 74% in *Hylobates*) are not shared between the two species and we now report this in the revised text (Lines 265-272). These findings reinforce the idea that chromosomal rearrangements and many of the TAD boundaries that overlap them, have emerged relatively recently. Lastly, we have now annotated the BOS overlapping with TAD boundaries in *Nomascus* and *Hylobates* that are “Ancestral” (i.e. shared) vs. “*Hylobates*-specific” or “*Nomascus*-specific along with the full list of BOS in all species, in a new supplementary table (**Supplementary Table 8**).

5. The authors report significant overlaps between TAD boundaries, TEs (especially SINEs and LTRs), and synteny breaks. Like TAD boundaries, these classes of TEs as well as chromosomal rearrangements are well documented to preferentially occur in gene-rich, GC-rich regions of the genome. It is unclear that the whole genome is an appropriate background here to test co-occurrence and conclude that a plausible causal relationship exists between these genomic structures: at least, the permutation should be adjusted for %GC.

The reviewer brings up an important point. We should however clarify that our investigation of TE content, which focused on the CTCF binding sites (i.e. CTCF ChIP-seq peaks) within TAD boundaries (not the whole TAD boundary), were carried out using the TEanalysis software (Lynch et al., 2015; [https://www.cell.com/fulltext/S2211-1247\(14\)01105-X](https://www.cell.com/fulltext/S2211-1247(14)01105-X)). This program, which is commonly used to test TE content of given genomic features, determines the enrichment (or depletion) of each family/class of TE relative to the genomic abundance of that TE family/class. Briefly, for each TE, the proportion of TE counts overlapping features (in this case CTCF peaks in TAD boundaries) is compared to the genomic abundance of that family (with 100% corresponding to the total amount of the same TEs in the genome). The ratio between counts (in-feature divided by in-genome) is used to estimate if a given TE is enriched or depleted, and significance is then inferred using the negative binomial test. Therefore, this analysis does not use manual permutations/randomizations.

We did however use random permutations for investigating BOS/boundary overlap, and we originally used simple random sampling (without controlling for GC content) since our findings recapitulated patterns of co-occurrence that have been reported several times previously. Because we agree that selecting an appropriate background is key when performing randomization permutations, we have repeated the analysis using stratified random sampling to control for the higher %GC of BOS regions when generating our random background. This adjustment did not change our main findings, but the boundary/BOS co-occurrence we previously reported in the rhesus genome is no longer significant ($p > 0.05$). We have updated the Methods (lines 565-569), Results (Lines 252-254) and Figure 3, accordingly.

6. I am confused by the first reported example of TAD boundary deletion, where the authors edited a 23 kb region near *Dmrtb1* in a mouse mutant. According to Fig 4a, that deletion does not cover the conserved TAD boundary, but actually targets a nearby CTCF site. I could not find evidence in the manuscript that any disruption of the conserved TAD boundary exists in this mutant – in fact, the authors report no differences in interaction frequencies at that border. The way the authors present this as evidence of the functional importance of ultra-conserved boundaries is highly misleading - it is more likely that the reported gene expression and phenotypic changes are due to the deletion of the CTCF site, as CTCF has many other gene regulatory roles besides its insulating properties.

We understand that both reviewers found this section confusing. We should clarify that the original design of the *B396_BOS* CRISPR deletion was informed by publicly available Hi-C data and CTCF ChIP-seq data from human, gibbon, rhesus and mouse, that we later replaced with the higher

resolution data presented here. We designed our CRISPR probes so they would delete a conserved CTCF ChIP-seq peak located within a conserved TAD boundary. After re-annotating the ultraconserved boundaries with the new high-resolution Hi-C data, we found a ~14kb distance between this deletion and the newly annotated ultraconserved boundary. In light of the critique from both reviewers we have now decided to remove the entire section describing this study (see gray and strikethrough text throughout manuscript). We believe that this decision does not impact the conclusions of our study since we demonstrate functionality of ultraconserved boundaries via deletion of another ultraconserved boundary in mouse model *B5234^{-/-}*.

7. I found the discussion a bit confusing, as the authors strongly emphasize that TAD boundaries likely “evolve under strong evolutionary constraints preventing their disruption”, which seems in direct contradiction with their findings that TAD boundaries are not very conserved and frequently remodeled by TEs.

The reviewer brings up a valid point. We have now changed the tone and language we use in several parts of the Discussion to better convey our interpretation of the results. To clarify, we think our findings show that a relatively small subset of boundaries display unprecedented cross-species conservation despite our strict criteria for conservation (e.g. the use of two tissues and only tolerating small (<10Kb) shifts in boundary position across species). The rest of boundaries display lower levels of conservation and some are only found exclusively in a single species. We agree that some boundaries may have been misclassified, and some may have been considered “conserved” if more lenient criteria was applied. Nevertheless, it is clear that TAD boundaries do vary in their level of their conservation, as is the case with any other functional genomic elements. Therefore, while there is some evolutionary conservation, we should not assume that all (or most) boundaries remain stable during evolution.

Minor points

- Figure 1 and 2a-d are largely redundant information. I would suggest to condense.

Both reviewers found these figures to be redundant. In the interest of conciseness we have now moved Fig 1b and 1c to Extended Figure 3.

- “Human LCL genome”, “mouse liver genome”: this is weird phrasing, consider modifying to “human genome in LCL”/“mouse genome in liver”, etc.

We have revised the text accordingly.

- “Deletion of ultraconserved TAD boundaries results in selective phenotypes”: that paragraph title is misleading, the authors do not test for selection here, only functional impact (functional changes can be non-selective)

We agree with this and have revised the title to say “Deletion of ultraconserved TAD boundaries results in tissue-specific functional phenotypes”

Reviewer #3 (Remarks to the Author):

Review of the manuscript:

“TAD evolutionary and functional characterization reveals diversity in mammalian TAD boundary properties and function”

How chromatin organization evolves at the level of TADs and its implication in the evolution of gene regulation is a highly interesting and debated topic. The manuscript of Okhovat et al. helps to shed light

on this question contributing valuable Hi-C and epigenetic datasets in different species and the analysis of functional experiments in disparate organs like testis and heart. I find the manuscript to be valuable for the community and the readership of Nature Communications. However, I have concerns and I disagree with some of the conclusions presented.

We would like to thank the reviewer for finding our work of value to the community and for their careful examination and constructive feedback. We have addressed their concerns below and we hope that they find our answers, new experiments and manuscript revisions satisfactory.

Major points

1. The authors claim that their TAD boundary conservation data challenge the assumption that TAD organization is conserved across tissues / evolution because of the low percentage (14-15%) of boundaries common in the eight different species. I would not interpret the data in that way, since also more than 50% of the TAD boundaries are conserved in at least 4 other species. It is expected that increasing the number of species used in the comparison also increases the chance that a particular boundary is not found in at least one of the other species. Besides, small differences in insulation can cause boundary calling algorithms to fluctuate specially in the “weak” boundaries regime. Furthermore, the conservation of TAD boundary locations is not necessarily the best way to address conservation of TAD structures (albeit it is perhaps the most straightforward). For instance, a boundary might be displaced 100 kb upstream or downstream in a particular species and count as not conserved. However, the content of the rest of the TAD (which might consist of 500 or 600 kb) in terms of regulatory regions and syntenic genes might be almost identical. If the attention is only focused on the boundary location, this TAD would be considered as entirely divergent even though 90% of its content is conserved. In summary, I believe that only focusing on the fact that “only” 14% of boundaries are conserved in 8 out of 8 species gives the false impression that 80% of the TAD structures in mammals are evolving almost neutrally. Which I think is a notion that the very same data presented in this study speak against.

We understand the reviewer’s concern and agree with their interpretation of our data. We have modified the text in our manuscript to clarify that our approach in detecting TAD conservation is fairly conservative and is primarily based on the exact placement of TAD boundaries, not TAD gene content. In other words, our list of ultraconserved boundaries represent boundaries that display very high levels of conservation in both strength and genomic placement across species. Given our strict criteria for conservation, and in light of constructive critique from both reviewers, we have now adjusted and softened the tone we use when describing the level of boundary conservation in the manuscript.

2. I am concerned that weaker boundaries might be more difficult to find by the boundary caller in other species and this analysis might contain false negatives. This will influence the conclusion that weaker boundaries are less conserved, maybe they are just closer to the limit of being called a boundary or not.

While our findings are in line with a previous study (Kentepozidou et al. 2020), this is a valid critique and it is certainly possible that weaker boundaries may be missed in some species. We now briefly mention this in the revised Discussion (see lines 414-416).

3. The authors claim that BOS-overlapping TAD boundaries were formed later or coordinated with the break of synteny event because they tend to occur at “young” boundaries. I think this is a very intriguing finding that should be discussed further: it challenges the interpretation that BOS happen at boundaries because then TAD structure is preserved. What force would create a boundary after a rearrangement?

We also find this observation interesting and in the revised manuscript we also note that most of the BOS associated with TAD boundaries in the two gibbon species are species-specific (see new text in lines 265-272), supporting a roughly similar evolutionary timeline for the formation of boundaries and chromosomal rearrangements. While we do not yet know the molecular forces that might have been involved in creating a boundary immediately after or at the same time as the chromosomal rearrangements, we speculate that co-option of transposable elements carrying CTCF sites might have been the most immediate way for introducing insulation and reducing possible ectopic chromatin contacts between newly joined chromosome regions. We have now add our speculation to the revised manuscript (see lines 441-448).

4. In the deletion of the CTCF site overlapping the gibbon BOS at the *Dmrt1b* locus, the nature of this BOS is not clarified. Is this an inversion, a translocation, an insertion? Where is the other breakpoint? Is the other breakpoint inside the same TAD or is the BOS altering the TAD structure? It is difficult to interpret the rearrangement effects without this information.

Please see our comment below.

5. The resolution of the Capture-C data from the *Dmrt1b* locus is insufficient to rule out whether there are effects at the level of chromatin organization. I think that at least some 4C-seq (ideally UMI) with sufficient resolution should be provided to clarify the effects of the mutation on the regulatory landscape of *Dmrt1b*. Also, the exact location of the boundary is unclear from the Capture-C data, and the Hi-C data is also not shown. The data suggest enhancer adoptions of *Lrp8* enhancers by *Dmrt1b* but with the current experiments is difficult to tell.

Considering the ambiguities brought up by reviewers (please also see concern #6 from Reviewer 1), we have removed the entire section describing generation and phenotyping of the *B396_BOS*^{-/-} mouse model, including the Capture-HiC data brought up here. Since we already have intriguing molecular/functional results from deletion of another candidate ultraconserved boundary, removing this section does not impact the integrity of our story. Hopefully this decision will improve the clarity of our study and address both reviewer's concerns.

That said, to clarify, we preferred to use Capture-Hi-C rather than 4C-seq, because we were interested in investigating all potential interaction changes around the deletion, not just those anchored at the gene of interest. We expect our assay to be able to detect gross conformation changes that reflect TAD reorganization. In fact, Capture-Hi-C (of the same resolution and coverage) on the *meis2* locus in the heart of *B5234*^{-/-} mice revealed changes in genome organization following deletion of the ultraconserved boundary. Nonetheless, we do agree that increasing resolution may reveal subtle interaction changes at the *Dmrtb1* locus, and based on our histology data we would expect these changes to be cell-type/stage specific.

6. Chromosome conformation capture data (either Hi-C, Capture-C or 4C-seq) should be provided also for mutants and controls in the *Meis2* and *AUTS2* locus. In the case of *AUTS2* Hi-Cs are available but only the insulation score is shown.

We have now performed Capture-HiC on the hearts of wild-type and KO *Meis2* mutant mice and added results from this new assay to the revised Results (see lines 336-339) and new panel in Figure 5 (included below for the reviewers consideration). As apparent from the Hi-C heatmap matrix and the insulation curves, deletion of the candidate ultraconserved boundary results in chromatin changes consistent with merging of neighboring TADs (i.e. increased interaction occurring across the deletion site and loss of the boundary insulation dip in mutant hearts).

We have also added the Hi-C heatmap for the *AUTS2* locus as a supplementary figure (**Supplementary Figure 6**).

Minor points

1. I find redundant the content of Fig. 2a-d and Fig. 1.

This same issue was brought up by Reviewer 1 and while there are nuanced differences in the nature of the data presented in these two figures, we do agree that the overall message is redundant. Therefore, we have now moved Fig 1b,c to Extended Fig. 3.

2. Fig. 2c do not seem to correspond to the statement “species-specific TAD boundaries in human and mouse show higher enrichment of chromatin states associated with active transcription start sites, bivalent chromatin and CTCF signal”. In the heatmaps it looks like the other way around.

We would like to thank the reviewer for spotting this typo. We have now corrected the text to say “species-specific TAD boundaries in human and mouse show **lower** enrichment of chromatin states associated with active transcription start sites, bivalent chromatin and CTCF signal”.

3. The labels asterisks and scRNA, tRNA labels of figure 2e are confusing and their meaning is not defined in the caption.

We have now clarified the meaning of labels and asterisks in the figure caption.

REVIEWERS' COMMENTS

Reviewer #1 (Remarks to the Author):

I thank the authors for their detailed response to my comments. The clarifications and modifications to the manuscript addressed all my major concerns.

I find that the PCA included in the response and Supp Fig 2 should possibly be moved either to the main or extended figures, as it does nicely show the moderate impact of tissue/cell of origin with regard to the phylogenetic distance (although I suppose this might be a little confounded by alignability, as well). This is very minor though and I will leave this to the authors and editors decision.

Reviewer #3 (Remarks to the Author):

I want to thank the authors for the revised version of the manuscript. My main concerns before this round of revision were:

- a) The interpretation of degree of conservation between TADs, which has been significantly reinterpreted by the authors. I think now this interpretation is more careful and accurate.
- b) The design of the Dmrt1b experiment was flawed and the experiments to characterize the mutants were insufficient. This section has been removed and I think the paper benefits from this decision. The Dmrt1b results were difficult to interpret and likely unrelated to TAD boundary function.
- c) 3D chromatin conformation data was missing to characterize the result of the Meis2 boundary mutation. Now they have been added and are compatible with the gene expression and the heart phenotypes described.
- d) Hi-C data was not shown for the AUTS2 locus, now it is available as a Supplementary Figure. I would nonetheless encourage the authors to show the deletion coordinates in that figure.

Therefore my main concerns have been addressed and I consider that the paper is ready for publication in Nature Communications.

REVIEWERS' COMMENTS

Reviewer #1 (Remarks to the Author):

I thank the authors for their detailed response to my comments. The clarifications and modifications to the manuscript addressed all my major concerns.

We are glad that our responses and modifications addressed the major concerns previously raised by this reviewer.

I find that the PCA included in the response and Supp Fig 2 should possibly be moved either to the main or extended figures, as it does nicely show the moderate impact of tissue/cell of origin with regard to the phylogenetic distance (although I suppose this might be a little confounded by alignability, as well). This is very minor though and I will leave this to the authors and editors decision.

We agree with the reviewer and we have now moved this figure to be part of the Extended Data Figure 1.

Reviewer #3 (Remarks to the Author):

I want to thank the authors for the revised version of the manuscript. My main concerns before this round of revision were:

a) The interpretation of degree of conservation between TADs, which has been significantly reinterpreted by the authors. I think now this interpretation is more careful and accurate.

b) The design of the Dmrt1b experiment was flawed and the experiments to characterize the mutants were insufficient. This section has been removed and I think the paper benefits from this decision. The Dmrt1b results were difficult to interpret and likely unrelated to TAD boundary function.

c) 3D chromatin conformation data was missing to characterize the result of the Meis2 boundary mutation. Now they have been added and are compatible with the gene expression and the heart phenotypes described.

d) Hi-C data was not shown for the AUTS2 locus, now it is available as a Supplementary Figure. I would nonetheless encourage the authors to show the deletion coordinates in that figure.

We have now revised the supplementary figure to mark the position of the deletion.

Therefore my main concerns have been addressed and I consider that the paper is ready for publication in Nature Communications.

We are glad that the main concerns have been addressed and that the reviewer deems the manuscript ready for publication.